# DICE: DATA INFLUENCE CASCADE IN DECENTRALIZED LEARNING

**Tongtian Zhu , Wenhao Li & Can Wang**
Zhejiang University
{raiden,wenhao-li,wcan}@zju.edu.cn

**Fengxiang He**
University of Edinburgh
F.He@ed.ac.uk

## ABSTRACT

Decentralized learning offers a promising approach to crowdsource data consumptions and computational workloads across geographically distributed compute interconnected through peer-to-peer networks, accommodating the exponentially increasing demands. However, proper incentives are still in absence, considerably discouraging participation. Our vision is that a fair incentive mechanism relies on fair attribution of contributions to participating nodes, which faces non-trivial challenges arising from the localized connections making influence "cascade" in a decentralized network. To overcome this, we design the first method to estimate **D**ata **I**nfluence **C**ascad**E** (DICE) in a decentralized environment. Theoretically, the framework derives tractable approximations of influence cascade over arbitrary neighbor hops, suggesting the influence cascade is determined by an interplay of data, communication topology, and the curvature of loss landscape. DICE also lays the foundations for applications including selecting suitable collaborators and identifying malicious behaviors. Project page is available at 🔗 DICE.

## 1 INTRODUCTION

Large language models (LLMs) have seen remarkable progress in recent years (Guo et al., 2025; OpenAI, 2024; Dubey et al., 2024; DeepMind, 2024; Anthropic, 2024), surpassing human on key benchmarks (Maslej et al., 2024). The compute scaling, highlighted by Ho et al. (2024) as a major reason of the successes, is estimated by Epoch AI to increase four to five times annually in cutting-edge models (Sevilla & Roldán, 2024). This dramatic computational demand requires substantial financial investments; for example, training OpenAI's GPT-4 requires approximately $78 million in compute costs (Maslej et al., 2024). Such exorbitant expenses are far beyond the affordability of most smaller players, making tech giants increasingly dominant.

Currently, large-scale training and inference processes are primarily performed in expensive data centers. Decentralized training, echoing swarm intelligence (Bonabeau et al., 1999; Mavrovouniotis et al., 2017), offers a cost-efficient alternative avenue by crowd-sourcing computational workload to decentralized compute nodes (Yuan et al., 2022; Jaghouar et al., 2024). One notable example showcasing decentralized computing's computational potential is the Bitcoin eco-system which virtually distributes jobs requiring instantaneous 16 GW power consumption (CCAF, 2023).

Despite profound potential advantages, contributing to decentralized training incurs non-negligible costs for participants, raising a natural question: *What motivates edge participants to engage in decentralized training?* Game theory suggests that when appropriate incentives exist, self-interested (rational) players can be keen to contribute for socially desirable outcomes. It is thus essential to design a proper incentive mechanism to unleash the collectively massive computational potential of decentralized nodes. Our vision is that such incentive mechanism relies on accurate quantification of contributions from players. This leads to the following problem:

> *How to quantify individual contributions in decentralized learning?*

Quantifying the contributions (or "influence") to the learned model has been well studied in the centralized paradigm (Koh & Liang, 2017a; Pruthi et al., 2020). However, it is still largely untouched in understanding and measuring data influence in fully decentralized environments. Unlike centralized

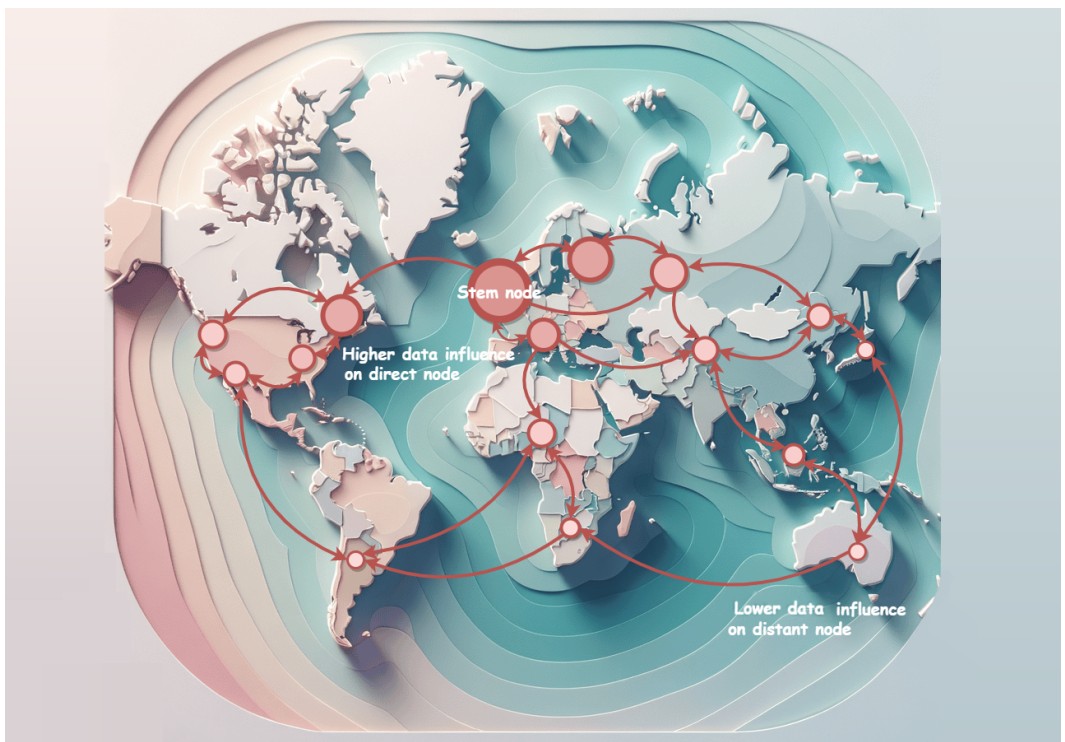

Figure 1: A semantic visualization of influence cascade in decentralized learning with ResNet-18 on CIFAR-10. The illustration depicts a 16-node communication topology (see Figure D.25), where node sizes represent DICE-E influence scores (see Theorem 2). Influence originates from the stem node and propagates through the network, weakening over distance, akin to "ripples in water". This highlights how data contribution extends beyond local nodes, shaped by the communication topology.

scenarios, where data influence is computed on a single model statically, decentralized learning relies on localized, indirect communications on a perhaps sparse network – the influence of data on one node impacts its own model, propagates to its neighbors through iterative parameter exchanges, and cascades to multi-hop neighbors. We term this mechanism as *cascading influence* (see Figure 1). Existing data influence estimators tailored for centralized settings assume that influence is computed within a single model and do not account for its recursive propagation through parameter exchange. As a result, they are not applicable to decentralized learning, where data influence extends beyond direct neighbors to multi-hop neighbors.

To address this challenge, we propose a **D**ata **I**nfluence **C**ascad**E** (DICE), the first work for measuring the influence in a decentralized learning environment. We make contributions as follows:

- **Conceptual contributions**: DICE introduces the concept of a ground-truth data influence for decentralized learning, integrating direct and indirect contributions to capture influence propagation across multiple hops during training (see Definition 3).

- **Theoretical contributions**: Building on this foundation, we derive tractable approximations of ground-truth DICE for an arbitrary number of neighbor hops, establishing a foundational framework to systematically characterize the flow of influence across decentralized networks. These theoretical results uncover, for the first time, that data influence in decentralized learning extends beyond the data itself and the local model, as seen in centralized training. Instead, it is a joint product of three critical factors: the original data, the topological importance of the data keeper, and the curvature information of intermediate nodes mediating propagation (see Theorem 2).

We anticipate that our DICE framework will pave the way for novel incentive mechanism designs and the establishment of economic opportunities for decentralized learning, such as data and parameter markets. DICE also holds significant potential to address critical challenges of identifying new suitable collaborators, and detecting free-riders. We envision these applications will contribute to a scalable, autonomous, and reciprocal decentralized learning eco-system.

## 2 RELATED WORK

**Data Influence Estimation.** As high-quality data becomes increasingly critical in modern machine learning (Hoffmann et al., 2022; Penedo et al., 2023; Li et al., 2023; Villalobos et al., 2024), understanding its influence has emerged as a key research direction (Sorscher et al., 2022; Grosse et al., 2023). Data influence estimation quantifies the contribution of training data to model predictions (Chen et al., 2024a; Ilyas et al., 2024), supporting incentive mechanisms and applications in few-shot learning (Park et al., 2021), dataset pruning (Yang et al., 2023), distillation (Loo et al., 2023), fairness (Li & Liu, 2022), machine unlearning (Sekhari et al., 2021), explainability (Koh & Liang, 2017b; Grosse et al., 2023), and security (Demontis et al., 2019; Hammoudeh & Lowd, 2022).

Existing methods fall into static and dynamic categories. Static approaches, including retraining-based methods (e.g., leave-one-out (Cook, 1977), Shapley value (Shapley, 1953), Datamodels (Ilyas et al., 2022)) and one-point methods (e.g., influence functions (Koh & Liang, 2017b)), estimate influence post-training. While theoretically grounded, these methods cannot characterize dynamic influence during training. Dynamic approaches address this limitation by tracking model parameter evolution (Charpiat et al., 2019). Notable methods include TracIn (Pruthi et al., 2020) and In-Run Data Shapley (Wang et al., 2024), which average gradient similarities over time. Recent advances (Nickl et al., 2023) leverage memory-perturbation equations to extend dynamic influence estimation to various optimization algorithms. For a more detailed background, please refer to Appendix A.1.

However, existing methods primarily focus on centralized training. To the best of our knowledge, the most closely related work is by Terashita & Hara (2022), who propose a decentralized hyper-gradient method and offer novel insights into using hyper-gradients to compute data influence. Nevertheless, their estimation method is static and cannot capture the influence cascade in decentralized training. In contrast, our framework, DICE, is specifically designed for fully decentralized environments, providing a fine-grained characterization of influence propagation unique to these settings.

**Incentivized Decentralized Learning**. Most existing incentive mechanisms for collaborative learning are designed for federated learning (Zeng et al., 2021). For instance, Wang et al. (2023b) propose an Incentive Collaboration Learning (ICL) framework to promote collaboration. Their focus is on mechanism design rather than the precise quantification of individual contributions. In federated learning, the Shapley value has been effectively utilized to quantify participant contributions (Jia et al., 2019; Ghorbani & Zou, 2019; Wang et al., 2019; 2020). Our approach differs fundamentally in two key aspects: first, we focus on fully decentralized settings without central servers, although our framework supports federated learning scenarios (see Algorithm 1); second, our work considers influence cascade between participants, an completely new perspective that has not been explored in existing literature. Regrading decentralized learning, we are only aware of the work by Yu et al. (2023) presenting a blockchain-based incentive mechanism for fully decentralized learning. However, their mechanism relies on smart contracts and differs from ours.

## 3 NOTATIONS AND PRELIMINARIES

This section introduces notations and essential preliminaries for decentralized learning. This work focuses on the most studied form of decentralized learning: data parallelism with only peer-level communication. For more detailed background, please refer to Appendix A.2.

We consider a general *personalized distributed optimization problem* over a connected graph $G = (\mathcal{V}, \mathcal{E})$, where $\mathcal{V}$ represents the set of participants and $\mathcal{E}$ denotes the communication links between them. The participants collaboratively minimize a weighted sum of local personalized objectives (T. Dinh et al., 2020; Hanzely & Richtárik, 2020):

$$\min_{\boldsymbol{\theta}=\{\boldsymbol{\theta}_k \in \mathbb{R}^d\}_{k \in \mathcal{V}}} \left[ L(\boldsymbol{\theta}) \triangleq \sum_{k \in \mathcal{V}} q_k L_k(\boldsymbol{\theta}_k) \right], \tag{1}$$

where $q_k \geq 0$ with $\sum_{k \in \mathcal{V}} q_k = 1$, and each local objective $L_k(\boldsymbol{\theta}_k) = \mathbb{E}_{\boldsymbol{z}_k \sim \mathcal{D}_k} [L(\boldsymbol{\theta}_k; \boldsymbol{z}_k)]$ is defined by the expectation over the local data distribution $\mathcal{D}_k$. Empirical risk minimization involves

optimizing the sample average approximation:

$$\hat{L}(\boldsymbol{\theta}) = \sum_{k \in \mathcal{V}} q_k \hat{L}_k(\boldsymbol{\theta}_k) \quad \text{where} \quad \hat{L}_k(\boldsymbol{\theta}_k) = \frac{1}{n_k} \sum_{i=1}^{n_k} L(\boldsymbol{\theta}_k; \boldsymbol{z}_{k_i}). \tag{2}$$

Here, $n_k$ is the number of samples in participant $k$, and $\{\boldsymbol{z}_{k_i}\}_{i=1}^{n_k}$ are drawn from $\mathcal{D}_k$.

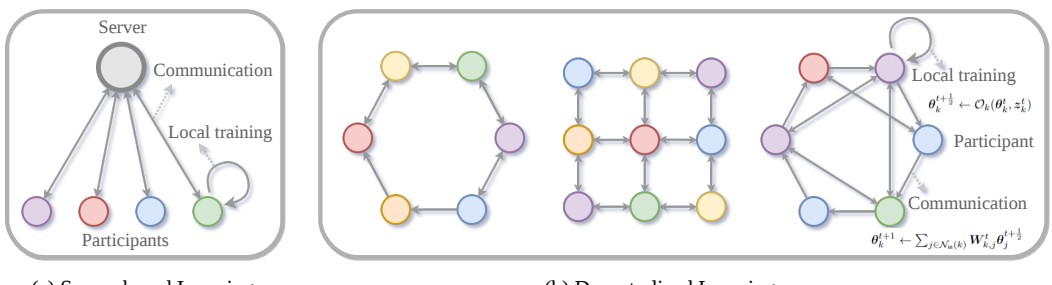

(a) Server-based Learning                   (b) Decentralized Learning

Figure 2: A comparative illustration of server-based learning versus decentralized learning.

Decentralized learning aims to minimize the global objective $l(\boldsymbol{\theta}) = \sum_{k \in \mathcal{V}} q_k l_k(\boldsymbol{\theta}_k)$ with only local computations and gossip communications among neighboring participants (Tsitsiklis et al., 1986; Nedic & Ozdaglar, 2009). The communication protocol is governed by a weighted adjacency matrix $\boldsymbol{W} \in [0,1]^{n \times n}$, where $\boldsymbol{W}_{k,j} \geq 0$ represents the strength of connection from participant $j$ to participant $k$, with $\boldsymbol{W}_{k,j} > 0$ if $(k,j) \in \mathcal{E}$. This matrix characterizes the communication topology, thereby defining how information propagates through the network (see Figure A.1). In this paper, $\boldsymbol{W}$ is designed to be row-stochastic, satisfying $\sum_{j=1}^{n} \boldsymbol{W}_{k,j} = 1$ for all $i \in \mathcal{V}$[1]. Decentralized learning alternates between local optimization and gossip-based parameter aggregation, as shown below:

---

**Algorithm 1** Decentralized Learning with Flexible Gossip and Optimization

---

**Require:** $G = (\mathcal{V}, \mathcal{E})$, $\{\boldsymbol{\theta}_k^0\}_{k \in \mathcal{V}}$, optimizer $\mathcal{O}_k$, number of communication rounds $T$, and mixing matrix distributions $\mathcal{W}^t$ ($\forall t \in [T]$)

1: **for** $t = 1$ to $T$ **do in parallel for all** participants $k \in \mathcal{V}$

2:      **Local Update:**

3:      Sample $\boldsymbol{z}_k^t \sim \mathcal{D}_k$, update parameters with optimizer $\mathcal{O}_k$: $\boldsymbol{\theta}_k^{t+\frac{1}{2}} \leftarrow \mathcal{O}_k(\boldsymbol{\theta}_k^t, \boldsymbol{z}_k^t)$

4:      **Gossip Averaging:**

5:      Send $\boldsymbol{\theta}_k^{t+\frac{1}{2}}$ to $\{l \mid \boldsymbol{W}_{l,k} > 0\}$ and receive $\boldsymbol{\theta}_j^{t+\frac{1}{2}}$ from $\{j \mid \boldsymbol{W}_{k,j} > 0\}$.

6:      Sample $\boldsymbol{W}^t \sim \mathcal{W}^t$, perform gossip averaging: $\boldsymbol{\theta}_k^{t+1} \leftarrow \sum_{j \in \mathcal{N}_{\text{in}}(k)} \boldsymbol{W}_{k,j}^t \boldsymbol{\theta}_j^{t+\frac{1}{2}}$

     **End for**

---

**Remark 1.** Algorithm 1 provides a flexible framework for decentralized learning with arbitrary optimizers and randomized gossip. A special case of this framework is decentralized stochastic gradient descent (DSGD) (Yuan et al., 2016a; Lian et al., 2017; Koloskova et al., 2020), where the optimizer $\mathcal{O}_k$ performs a simple stochastic gradient step: $\boldsymbol{\theta}_k^{t+\frac{1}{2}} \leftarrow \boldsymbol{\theta}_k^t - \eta^t \nabla L(\boldsymbol{\theta}_k^t; \boldsymbol{z}_k^t)$, with $\eta^t$ being the learning rate. Another notable special case is FedAVG (McMahan et al., 2017), which corresponds to the standard server-based learning setting, where a central server collects and averages model updates from all participants in each round. Mathematically, this is equivalent to using a fully connected and uniform mixing matrix in Algorithm 1, i.e., $\boldsymbol{W}_{k,j}^t \equiv \frac{1}{n} \mathbf{1}\mathbf{1}^T$ (see Figure 2 for a visual comparison and its connection to decentralized learning). Therefore, the framework is applicable to both federated and decentralized learning paradigms, although our primary focus remains on fully decentralized learning without a central server.

---

[1]The weighted adjacency matrix $\boldsymbol{W}$ is typically assumed to be doubly-stochastic. The row-stochastic assumption is weaker, yet the convergence of decentralized SGD is still guaranteed (Yuan et al., 2019; Xin et al., 2019).

## 4  DATA INFLUENCE CASCADES

In this section, we introduce DICE, a comprehensive framework for measuring data influence in decentralized environments. Subsection 4.1 introduces the ground-truth influence measures designed for decentralized learning and Subsection 4.2 provides their dynamic gradient-based estimations.

### 4.1  GROUND-TRUTH INFLUENCE IN DECENTRALIZED LEARNING

To ensure a logical and coherent flow, we first introduce the fundamental concepts of data influence in centralized settings and then discuss the significant challenges involved in extending these ideas to decentralized environments. In conventional centralized setups, the influence of an individual data instance can be assessed by evaluating the counterfactual change in learning performance through leave-one-out retraining (LOO) (Cook, 1977), defined as follows:

**Definition 1** (Leave-one-out Influence)**.**

$$\mathcal{I}_{\text{LOO}}(\boldsymbol{z}, \boldsymbol{z}') = L(\boldsymbol{\theta}^*; \boldsymbol{z}') - L(\boldsymbol{\theta}^*_{\setminus \boldsymbol{z}}; \boldsymbol{z}'), \qquad (3)$$

where $\boldsymbol{z}$ denotes the training data instance under influence assessment, $\boldsymbol{z}'$ is the loss-evaluating instance, $\boldsymbol{\theta}^*$ and $\boldsymbol{\theta}^*_{\setminus \boldsymbol{z}}$ are the models trained on the entire dataset $\mathcal{S}$ and $\mathcal{S} \setminus \{z\}$, respectively.

Intuitively, Equation (3) quantifies the influence of $\boldsymbol{z}$ by its individual impact on test loss reduction. A smaller LOO value indicates a significant contribution to learning, which aligns with the concept that the data influence is reflected in its ability to enhance model performance. LOO influence is often considered as the "gold standard" for evaluating how well influence estimators approximate the ground-truth influence in the data influence literature (Koh & Liang, 2017b; Basu et al., 2021).

However, extending LOO to decentralized scenarios introduces non-trivial challenges due to the distributed nature and the localized connections in decentralized learning, reflected in Equation (2) and Algorithm 1. In centralized setups, the core idea of LOO is to link data influence to variations in loss or parameter outcomes. In contrast, decentralized learning systems involve multiple participants sharing model parameters through inter-participant communications. As a result, alterations in model parameters caused by a data-level modification propagate throughout the whole network.

A natural way to measure the influence of one participant in such collaborative environments is through evaluating its contribution to the whole community (Wang et al., 2020; Yu et al., 2020), which aligns with the customer-centric principle (Drucker, 1985) in determining value[2]. In decentralized learning, when a participant transmits its training assets (e.g., model parameters or gradients) to neighboring participants—akin to offering a product—the recipients derive possible utility from these training assets and may provide reciprocal feedback, such as sharing their own assets in return. This dynamic positions the neighbors as "customers", thereby entrusting them with the rights to determine the value of the assets provided by the supplier. With these insights in mind, we recognize that assessing data influence in decentralized scenarios is far more complex, as summarized below:

> **Key observations**: *In decentralized learning,*
> *1) neighbors who serves as customers hold the rights to determine data influence;*
> *2) data influence is not static but spreads across participants through gossips during training.*

Unfortunately, existing static estimators only calculate the loss change after training and thus cannot characterize the dynamic transmission of data influence within the whole decentralized learning community. Based on the above discussion, we posit that a "gold-standard" influence measure in decentralized scenarios should satisfy the following requirements:

- Quantify community-level influence: Measure the impact of training data instances on the collective utility of the community.

- Depend on training dynamics: Measure the influence based on the training process to characterize the propagation of influence on decentralized networks.

---

[2]Peter Drucker's customer-centric value principle from *Innovation and Entrepreneurship* states, "Quality in a product or service is not what the supplier puts in. It is what the customer gets out of it.".

In the following, we introduce the ground-truth influence measures tailored to the requirements of decentralized environments, termed as the *ground-truth influence cascade (DICE-GT)*.

**Definition 2** (One-hop Ground-truth Influence). The one-hop DICE-GT value quantifies the influence of a data instance $z_j^t$ from participant $j$ on a loss-evaluating instance $z'$ within itself and its immediate neighbors. Formally, for a given participant $j \in \mathcal{V}$:

$$\mathcal{I}_{\text{DICE-GT}}^{(1)}(z_j^t, z') = \underbrace{q_j \left( L(\theta_j^{t+\frac{1}{2}}; z') - L(\theta_j^t; z') \right)}_{\text{direct marginal contribution of } z_j^t \text{ to } j} + \underbrace{\sum_{k \in \mathcal{N}_{\text{out}}^{(1)}(j)} q_k \left( L(\theta_k^{t+1}; z') - L(\theta_{k \setminus z_j^t}^{t+1}; z') \right)}_{\text{indirect marginal contribution of } z_j^t \text{ to one-hop neighbors}},$$

where $\theta_j^{t+\frac{1}{2}}$ denotes the updated model parameters of $j$ after training on $z_j^t$ at iteration $t$ (see Algorithm 1). For each one-hop out-neighbor $k \in \mathcal{N}_{\text{out}}^{(1)}(j)$, $\theta_k^{t+1}$ denotes the averaged model parameters after receiving updated parameters $\{\theta_l^{t+\frac{1}{2}} | W_{k,l} > 0\}$ influenced by $z_j^t$, while $\theta_{k \setminus z_j^t}^{t+1}$ represents the model parameters of $k$ without the influence from $z_j^t$, i.e., $\theta_{k \setminus z_j^t}^{t+1} = \sum_{l \in \mathcal{N}_{\text{out}}(k) \setminus j} W_{k,l}^t \theta_l^{t+\frac{1}{2}} + W_{k,j}^t \theta_l^t$.

The economic intuition behind the DICE-GT value is that it captures both the direct marginal contribution of a data instance to itself and its subsequent impact on immediate neighbors. Specially, the first term $L(z'; \theta_j^{t+\frac{1}{2}}) - L(z'; \theta_j^t)$ captures the inter-node direct influence of training data instance $z_j^t$ on the test loss change at node $j$, which corresponds to the TracInIdeal influence in (Pruthi et al., 2020) designed for centralized scenarios. The second term $\sum_{k \in \mathcal{N}_{\text{out}}^{(1)}(j)} (L(z'; \theta_k^{t+1}) - L(z'; \theta_{k \setminus z_j^t}^t))$ measures the intra-node influence unique in decentralized learning, which aggregates the indirect influences on all one-hop neighbors, i.e., direct neighbors, of node $j$.

In decentralized learning environments, data influence propagates not only to immediate neighbors but also to multi-hop neighbors through the communication topology. To characterize this multi-hop influence, we extend the ground-truth influence cascade measure to arbitrary $r$-hop neighbors.

**Definition 3** (Multi-hop Ground-truth Influence). The multi-hop DICE-GT value quantifies the cumulative influence of a data instance $z$ on a loss-evaluating instance $z'$ across all nodes within $r$-hop neighborhoods of participant $j$. Formally, for a given participant $j \in \mathcal{V}$:

$$\mathcal{I}_{\text{DICE-GT}}^{(r)}(z_j^t, z') = q_j \left( L(\theta_j^{t+\frac{1}{2}}; z') - L(\theta_j^t; z') \right) + \sum_{s=1}^{r} \sum_{k \in \mathcal{N}_{\text{out}}^{(s)}(j)} q_k \left( L(\theta_k^{t+s}; z') - L(\theta_{k \setminus z_j^t}^{t+s}; z') \right),$$

where $\mathcal{N}_{\text{out}}^{(s)}(j)$ denotes the set of $s$-hop out-neighbors of $j$ (please refer to Appendix A.3 for details of high-order neighbors). Here $\theta_k^{t+s}$ and $\theta_{k \setminus z_j^t}^{t+s}$ represents the parameters of node $k$ at iteration $t + s$ when the influence from $z_j^t$ are included and excluded, respectively.

Analogous to Definition 2, the first term captures the direct influence of data $z_j^t$ on the loss at node $j$. The subsequent summation aggregates the indirect influences on all multi-hop neighbors up to $r$ steps away from node $j$[3]. The reason to measure test loss change at the $t + s$ step is that the impact of $z_j^t$ propagating to $k \in \mathcal{N}_{\text{out}}^{(s)}(j)$ requires $s$ steps. This layered formulation accounts for the multi-hop cascading effects through the network up to the specified order $r$.

## 4.2 DYNAMIC GRADIENT-BASED ESTIMATIONS

To meet the second aforementioned requirement of decentralized learning, we design dynamic gradient-based estimators for DICE-GT, called the *influence cascade estimations (DICE-E)*.

---

[3]We note that the future effect of $z_j^t$ on its local model $\theta_j^{t+s}$ over $s = 2, \ldots, r$ iterations is included within the indirect influence terms, as a node is inherently considered part of its own arbitrary-order neighbor.

**Proposition 1** (Approximation of One-hop DICE-GT). The one-hop DICE-GT value (see Definition 2) can be linearly approximated as follow:

$$\mathcal{I}^{(1)}_{\text{DICE-E}}(\boldsymbol{z}^t_j, \boldsymbol{z}') = -q_j \nabla L(\boldsymbol{\theta}^t_j; \boldsymbol{z}')^\top \Delta_j(\boldsymbol{\theta}^t_j, \boldsymbol{z}^t_j) - \sum_{k \in \mathcal{N}^{(1)}_{\text{out}}(j)} q_k \boldsymbol{W}^t_{k,j} \nabla L(\boldsymbol{\theta}^{t+1}_k; \boldsymbol{z}')^\top \Delta_j(\boldsymbol{\theta}^t_j, \boldsymbol{z}^t_j),$$

where $\Delta_j(\boldsymbol{\theta}^t_j, \boldsymbol{z}^t_j) = \mathcal{O}_j(\boldsymbol{\theta}^t_j, \boldsymbol{z}^t_j) - \boldsymbol{\theta}^t_j$. The proof is included in Appendix C.1.

**Additivity**. The one-hop DICE-E influence measure is additive over training instances. Specifically, for a mini-batch $\mathcal{B}^t_j$ from participant $j$, the total influence is the sum of individual influences:

$$\mathcal{I}^{(1)}_{\text{DICE-E}}(\mathcal{B}^t_j, \boldsymbol{z}') = \sum_{\boldsymbol{z}^t_j \in \mathcal{B}^t_j} \mathcal{I}^{(1)}_{\text{DICE-E}}(\boldsymbol{z}^t_j, \boldsymbol{z}'). \tag{4}$$

The additivity provides guarantees for efficient computation of DICE-E score for large mini-batches.

We can then extend the influence approximation to multi-hop neighbors in decentralized learning and show how the influence of a data instance cascades over the decentralized learning network.

**Theorem 2** (Approximation of $r$-hop DICE-GT). The $r$-hop DICE-GT influence $\mathcal{I}^{(r)}_{\text{DICE-GT}}(\boldsymbol{z}^t_j, \boldsymbol{z}')$ (see Definition 3) can be approximated as follows:

$$\mathcal{I}^{(r)}_{\text{DICE-E}}(\boldsymbol{z}^t_j, \boldsymbol{z}') = -\sum_{\rho=0}^{r} \sum_{(k_1,\ldots,k_\rho) \in P^{(\rho)}_j} \eta^t q_{k_\rho} \underbrace{\left(\prod_{s=1}^{\rho} \boldsymbol{W}^{t+s-1}_{k_s, k_{s-1}}\right)}_{\text{communication graph-related term}} \underbrace{\nabla L\left(\boldsymbol{\theta}^{t+\rho}_{k_\rho}; \boldsymbol{z}'\right)^\top}_{\text{test gradient}}$$

$$\times \underbrace{\left(\prod_{s=2}^{\rho} \left(\boldsymbol{I} - \eta^{t+s-1} \boldsymbol{H}(\boldsymbol{\theta}^{t+s-1}_{k_s}; \boldsymbol{z}^{t+s-1}_{k_s})\right)\right)}_{\text{curvature-related term}} \underbrace{\Delta_j(\boldsymbol{\theta}^t_j, \boldsymbol{z}^t_j)}_{\text{optimization-related term}}.$$

where $\Delta_j(\boldsymbol{\theta}^t_j, \boldsymbol{z}^t_j) = \mathcal{O}_j(\boldsymbol{\theta}^t_j, \boldsymbol{z}^t_j) - \boldsymbol{\theta}^t_j$, $k_0 = j$. Here $P^{(\rho)}_j$ denotes the set of all sequences $(k_1, \ldots, k_\rho)$ such that $k_s \in \mathcal{N}^{(1)}_{\text{out}}(k_{s-1})$ for $s = 1, \ldots, \rho$ (see Definition A.7) and $\boldsymbol{H}(\boldsymbol{\theta}^{t+s}_{k_s}; \boldsymbol{z}^{t+s}_{k_s})$ is the Hessian matrix of $L$ with respect to $\boldsymbol{\theta}$ evaluated at $\boldsymbol{\theta}^{t+s}_{k_s}$ and data $\boldsymbol{z}^{t+s}_{k_s}$. For the cases when $\rho = 0$ and $\rho = 1$, the relevant product expressions are defined as identity matrices, thereby ensuring that the r-hop DICE-E remains well-defined. Full proof is deferred to Appendix C.3.

Multi-hop DICE-E characterizes the cascading effects of data influence through multiple "layers" of communication. In this context, the influence of a data instance from participant $j$ can propagate through a sequence of intermediate nodes, reaching participants that are $\rho$ hops away.

Influence Dynamics: Exponential Decay and Topological Dependency. Theorem 2 demonstrates that the multi-hop influence of a data instance $\boldsymbol{z}j^t$ is governed by the product of communication weights $\prod s = 1^\rho \boldsymbol{W}k_s, ks - 1^{t+s-1}$ and Hessian-related terms $\prod_{s=2}^{\rho}(\boldsymbol{I} - \eta^{t+s-1} \boldsymbol{H}k_s^{t+s-1})$. This indicates that data influence in decentralized learning depends on the curvature of intermediate nodes and decays exponentially with each additional hop. Nodes with higher topological importance (e.g., node $j$ with large $\sum j = 1^n \boldsymbol{W}_{j,k}$) propagate their data influence more widely and with greater impact on global utility (see Figure 1). These characteristics underscore the interplay between the original data, the loss landscape curvature, and communication topology in shaping data influence.

## 4.3 PRACTICAL APPLICATIONS

In idealized scenarios, participants may seek to estimate the influence of their high-order neighbors on their local utility improvement. In practice, one-hop DICE-E emerges as a more suitable choice

due to its computational efficiency. Based on Proposition 1, we derive the peer-level contribution, which we refer to as the *proximal influence*.

**Definition 4** (Proximal Influence). The proximal influence of a data instance $\boldsymbol{z}_j^t$ from participant $j$ on participant $k$ at iteration $t$ is defined as follows:

$$\mathcal{I}_{\text{DICE-E}}^{k,j}(\boldsymbol{z}_j^t, \boldsymbol{z}') = -\eta^t \boldsymbol{W}_{k,j}^t q_k \nabla L(\boldsymbol{\theta}_j^t; \boldsymbol{z}_j^t)^\top \nabla L(\boldsymbol{\theta}_k^{t+1}; \boldsymbol{z}'). \tag{5}$$

This term quantifies the influence of the data instance $\boldsymbol{z}_j^t$ from participant $j$ on the loss reduction experienced by its immediate neighbor $k$. Importantly, under the information sharing protocol defined in Algorithm 1, participant $k$ has access to $q_k$, $\boldsymbol{W}_{k,j}^t$, $\nabla L(\boldsymbol{\theta}_j^t; \boldsymbol{z}_j^t)$, and $\nabla L(\boldsymbol{\theta}_k^{t+1}; \boldsymbol{z}')$.[4] Therefore, each participant can compute the proximal contributions of its neighbors. The proximal influence can be utilized in the following scenarios:

**Collaborator Selection**. In decentralized learning, local data remains private and only local parameter communication is permitted. The absence of a central authority complicates the problem of selecting the most suitable neighbors with high-quality data. Fortunately, DICE offers a mechanism for participants to efficiently estimate the contributions of their neighbors with proximal influence. By assessing the proximal influence of their neighbors, participants can identify the potential collaborators that have the most significant positive impact on their learning process.

To ensure reciprocal collaboration (Gouldner, 1960; Fehr & Gächter, 2000; Sundararajan & Krichene, 2023), participants can compute *reciprocity factors*, which evaluate the mutual balance of influence.

**Definition 5** (Reciprocity Factors). The *reciprocity factor* is defined in two forms:

1. **Proximal Reciprocity Factor:** The reciprocity factor between participants $j$ and $k$ at iteration $t$ is

$$R_{k,j}^t = \frac{q_k \boldsymbol{W}_{k,j}^t \nabla L(\boldsymbol{\theta}_k^{t+1}; \boldsymbol{z}')^\top \nabla L(\boldsymbol{\theta}_j^t; \boldsymbol{z}_k^t)}{q_j \boldsymbol{W}_{j,k}^t \nabla L(\boldsymbol{\theta}_j^{t+1}; \boldsymbol{z}')^\top \nabla L(\boldsymbol{\theta}_k^t; \boldsymbol{z}_j^t)}. \tag{6}$$

2. **Neighborhood Reciprocity Factor:** To evaluate reciprocity at the community level, the neighborhood reciprocity factor for participant $j$ at iteration $t$ is defined as:

$$R_j^t = \frac{\sum_{k \in \mathcal{N}_{\text{out}}^{(1)}(j)} q_k \boldsymbol{W}_{k,j}^t \nabla L(\boldsymbol{\theta}_k^{t+1}; \boldsymbol{z}')^\top \nabla L(\boldsymbol{\theta}_j^t; \boldsymbol{z}_j^t)}{\sum_{l \in \mathcal{N}_{\text{in}}^{(1)}(j)} q_l \boldsymbol{W}_{j,l}^t \nabla L(\boldsymbol{\theta}_j^{t+1}; \boldsymbol{z}')^\top \nabla L(\boldsymbol{\theta}_l^t; \boldsymbol{z}_j^t)}. \tag{7}$$

The proximal reciprocity factor measures the balance of influence between two participants, with values near unity indicating equitable mutual contributions. Significant deviations suggest an imbalance, helping participants refine their collaboration strategies. The neighborhood reciprocity factor extends this concept to a participant's local community, evaluating the balance between influence inflow and outflow. These metrics support participants in adjusting their engagement and aids the community in managing membership, such as admitting new members or excluding underperforming participants.

## 5 EXPERIMENTS

This section presents the experimental results, with implementation details outlined in Appendix D.1.

**Influence Alignment** We evaluate the alignment between one-hop DICE-GT (see Definition 2) and its first-order approximation, one-hop DICE-E (see Proposition 1). One-hop DICE-E $\mathcal{I}_{\text{DICE-E}}^{(1)}(\mathcal{B}_j^t, \boldsymbol{z}')$ is computed as the sum of one-sample DICE-E within the mini-batch $\mathcal{B}_j^t$ thanks to the additivity (see Equation (4)). DICE-GT $\mathcal{I}_{\text{DICE-GT}^{(1)}}(\mathcal{B}_j^t, \boldsymbol{z}')$ is calculated by measuring the loss reduction after removing $\mathcal{B}_j^t$ from node $j$ at the $t$-th iteration. As shown in Figure 3, each plot contains 30 points,

---

[4]In decentralized learning literature, it is common for each participant to share only local parameters with its neighboring participants (Lian et al., 2017; Koloskova et al., 2020). We note that sharing local gradients with neighbors maintains the decentralized learning paradigm and does not significantly compromise privacy, as a participant can reconstruct the gradients of its neighbors using the shared parameters (Mrini et al., 2024).

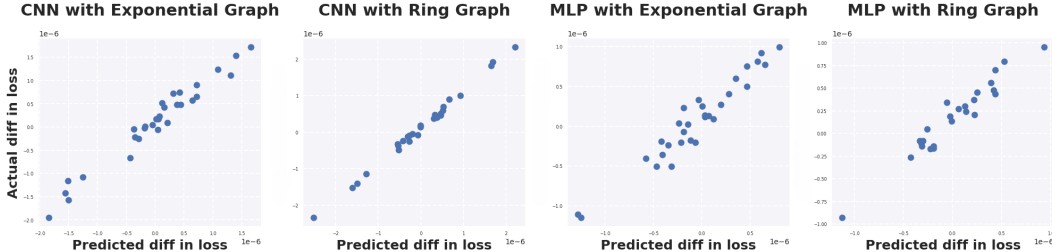

Figure 3: Alignment between one-hop DICE-GT (vertical axis) and DICE-E (horizontal axis) on a 32-node ring graph. Each node uses a 512-sample subset of Tiny ImageNet. Models are trained for 5 epochs with a batch size of 128 and a learning rate of 0.1.

with each point representing the result of a single comparison of the ground-truth and estimated influence. We can observe from Figure 3 that DICE-E closely tracks DICE-GT under different settings. The alignment becomes even stronger on simpler data set including CIFAR-10 and CIFAR-100, as detailed in Appendix D.2. These results demonstrate that DICE-E provides a strong approximation of DICE-GT, achieving consistent alignment across datasets (CIFAR-10, CIFAR-100 and Tiny ImageNet) and model architectures (CNN and MLP). Further validation of this alignment is provided in Appendix D.2 to corroborate the robustness of one-hop DICE-E under changing batch sizes, learning rates, and training epochs.

**Anomaly Detection**    DICE identifies malicious neighbors, referred to as anomalies, by evaluating their proximal influence, which estimates the reduction in test loss caused by a single neighbor. A high proximal influence score indicates that a neighbor increases the test loss, negatively impacting the learning process. In our setup, anomalies are generated through random label flipping or by adding random Gaussian noise to features, please kindly refer to (Zhang et al., 2024). Figure 3 illustrates that the most anomalies (in red) are readily detectable with proximal influence values. Additional results in Appendix D.3 further validate the reliability of this approach.

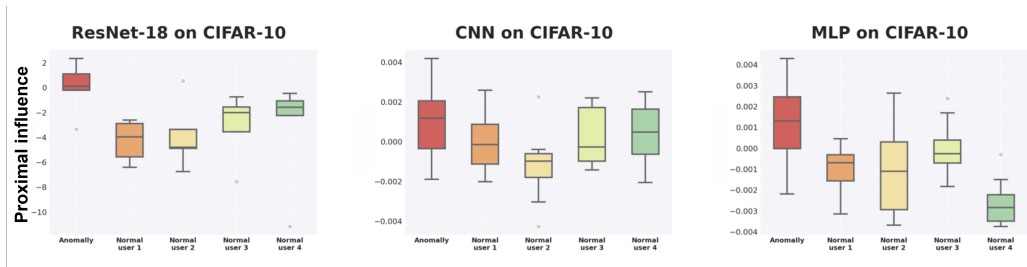

Figure 4: Anomaly detection on exponential graph with 32 nodes. Each node uses a 512-sample subset of CIFAR-10. Models are trained for 5 epochs with a batch size of 128 and a learning rate of 0.1. In a 32-node exponential graph, each participant connects with 5 neighbors, where the neighbor in red is set as an anomaly by random label flipping, while the other four are normal participants.

**Influence Cascade**    The topological dependency of DICE-E in our theory reveals the "power asymmetries" (Blau, 1964; Magee & Galinsky, 2008) in decentralized learning. To support the theoretical finding, we examine the one-hop DICE-E values of the same batch on participants with vastly different topological importance. Figure 1 illustrates the one-hop DICE-E influence scores of an **identical data batch** across participants during decentralized training of a ResNet-18 model on the CIFAR-10 dataset. Node sizes represent the one-hop DICE-E influence scores, quantifying how a single batch impacts other participants in the network. The dominant nodes (e.g., those with larger outgoing communication weights in $W$) exhibit significantly higher influence, as shown in Appendix D.4. These visualizations underscore the critical role of topological properties in shaping data influence in decentralized learning, demonstrating how the structure of the communication matrix $W$ determines the asymmetries in influence.

## 6 CONCLUSION AND FUTURE WORK

In this paper, we introduce DICE, the first comprehensive framework for quantifying data influence in fully decentralized learning environments. By modeling influence propagation across multiple hops, DICE reveals how local data contributions extend beyond immediate neighbors to reach non-adjacent neighbors. Mathematically, DICE formalizes how data influence cascades through the communication network, uncovering for the first time the intricate interplay between original data, communication topology, and the curvature of the optimization landscape in shaping data influence.

**Future Work**. Beyond its theoretical contributions, DICE holds significant potential for practical applications. By identifying influential contributors, DICE provides a foundation for designing fair incentive schemes that ensure equitable attribution of contributions, thereby promoting broader participation in decentralized learning. Moreover, DICE could contribute to the development of decentralized markets, including data markets (Huang et al., 2023), parameter markets (Fallah et al., 2024), and compute markets (Kristensen et al., 2024), which serve as fundamental building blocks of a broader decentralized learning ecosystem.

## ACKNOWLEDGMENT

This work is supported by the National Natural Science Foundation of China (No. 62476244), ZJU-China Unicom Digital Security Joint Laboratory and the advanced computing resources provided by the Supercomputing Center of Hangzhou City University.

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

# A    BACKGROUND

## A.1    DATA INFLUENCE

**Data Influence Estimation**. As high-quality data becomes increasingly critical in modern machine learning (Hoffmann et al., 2022; Penedo et al., 2023; Li et al., 2023; Longpre et al., 2024; Villalobos et al., 2024), understanding the influence of data has emerged as a crucial research direction (Sorscher et al., 2022; Grosse et al., 2023; Xia et al., 2024). Data influence estimation quantifies the contribution of training data to model predictions (Chen et al., 2024a; Ilyas et al., 2024). It enables fair attribution of data source contributions, serving as the foundation for incentive mechanisms. Besides serving as an incentive, data influence has been extensively applied in various machine learning domains, including few-shot learning (Park et al., 2021), dataset pruning (Sorscher et al., 2022; Yang et al., 2023), distillation (Loo et al., 2023), fairness improving (Li & Liu, 2022), machine unlearning (Guo et al., 2020; Sekhari et al., 2021), explainability (Koh & Liang, 2017b; Han et al., 2020; Grosse et al., 2023), as well as training-set attacks (Demontis et al., 2019; Jagielski et al., 2021) and defenses (Hammoudeh & Lowd, 2022).

Data influence estimators are broadly categorized into static and dynamic approaches[5]. Specifically, static approaches include both retraining-based and one-point methods. Retraining-based methods, such as leave-one-out (Cook, 1977), Shapley value (Shapley, 1953), and Datamodels (Ilyas et al., 2022), assess data influence by retraining the model on (subsets of) the training data. These methods offer a conceptually straightforward computation of data influence and are grounded in theoretical foundations, but they are often computationally expensive due to the requirement for retraining.

In contrast, one-point influence methods approximate the effect of retraining using a single trained model. A well-established one-point method is the canonical influence function (Koh & Liang, 2017b), developed from statistics (Hampel, 1974; Chatterjee et al., 1982), which examines how infinitesimal perturbations of a training example affect the empirical risk minimizers (ERM) (Vapnik & Chervonenkis, 1974). The influence function has been extended to incorporate higher-order information (Basu et al., 2020) and scaled for larger models (Guo et al., 2021; Schioppa et al., 2022), including LLMs (Grosse et al., 2023). Fu et al. (2022) extend influence function to Bayesian inference. While these static influence measures have elegant theoretical foundations, they are limited in characterizing how training data influences the training process.

Alternatively, dynamic methods enhance influence estimation by considering the evolution of model parameters across training iterations (Charpiat et al., 2019). Notable examples in this category include TracIn (Pruthi et al., 2020) and In-Run Data Shapley (Wang et al., 2024), which track the influence of training data points by averaging gradient similarities over time. The practicality of dynamic influence estimators is demonstrated by their applications in improving training processes in modern setups (Xia et al., 2024). Recently, Nickl et al. (2023) adopt a novel memory-perturbation equation framework to derive dynamic influence estimation of model trained under different centralized optimization algorithms, including SGD, RMSprop and Adam.

However, the existing static and dynamic influence estimation methods primarily target centralized scenarios, and little progress has been made in analyzing data influence in fully decentralized environments. To the best of our knowledge, the only closely related work is by Terashita & Hara (2022), who proposed a decentralized hyper-gradient method and provided novel insights on applying hyper-gradients to compute a centralized formulation of data influence. Nevertheless, their estimation method is static and cannot capture the influence cascade arising from gossip communication during decentralized training. In contrast, our framework, DICE, is specifically designed for fully decentralized environments, allowing it to provide a fine-grained characterization of the unique influence cascade inherent in these settings.

## A.2    DECENTRALIZED LEARNING

Currently, large-scale training and inference processes are primarily performed in expensive data centers. Decentralized training, echoing swarm intelligence (Bonabeau et al., 1999; Mavrovouniotis

---

[5]Typically, data influence estimators are classified as retraining-based and gradient-based methods (Hammoudeh & Lowd, 2024). For enhanced logical coherence in this paper, we group retraining-based methods and one-point gradient-based techniques under the static category. This classification does not contradict conventional categorizations.

et al., 2017), offers a cost-efficient alternative avenue by crowd-sourcing computational workload to geographically decentralized compute nodes, without the control of central servers (Yuan et al., 2022; Borzunov et al., 2023b; Jaghouar et al., 2024). One notable example showcasing decentralized computing's computational potential is the Bitcoin eco-system which virtually distributes jobs requiring instantaneous 16 GW power consumption (CCAF, 2023)– this has been triple of the estimated 5 GW of the world's largest planned cluster for AI (Gardizy & Efrati, 2024; OpenAI, 2025).

In the following, we provide an overview of the algorithmic and theoretical advancements in decentralized learning. While our discussion touches on several notable contributions, it is far from exhaustive. For a more comprehensive survey, we refer readers to Martínez Beltrán et al. (2023); Singha et al. (2024); Yuan et al. (2024).

**Algorithmic Development of Decentralized Learning**. The advancement of decentralized learning algorithms has been driven by the need for communication-efficient optimization methods in practical distributed learning scenarios. These algorithms have adapted to accommodate dynamic network structures (Nedi'c & Olshevsky, 2014; Koloskova et al., 2020; Ying et al., 2021; Takezawa et al., 2023), asynchronous communication (Lian et al., 2018; Xu et al., 2021; Nadiradze et al., 2021; Bornstein et al., 2023; Even et al., 2024), data heterogeneity (Tang et al., 2018; Vogels et al., 2021; Le Bars et al., 2023), and Byzantine adversaries (He et al., 2022; Ye & Ling, 2025). Furthermore, their applicability has extended beyond conventional optimization problem to more complex problem formulations, including compositional (Gao & Huang, 2021), minimax (Xian et al., 2021; Zhu et al., 2023a; Chen et al., 2024b), and bi-level (Yang et al., 2022; Gao et al., 2023; Chen et al., 2023) optimization problems. Additionally, privacy concerns in decentralized learning are also critical, with efforts focusing on differentially privacy (Cyffers et al., 2024; Allouah et al., 2024) and data reconstruction attacks (Mrini et al., 2024).

**Theoretical Development of Decentralized Learning**. In terms of optimization, earlier works on decentralized optimization (Nedic & Ozdaglar, 2009; Sayed, 2014; Yuan et al., 2016b; Lian et al., 2017) lay the groundwork for understanding convergence. Lu & De Sa (2021) present a systematic framework for federated and decentralized learning by categorizing decentralization into three distinct layers. Koloskova et al. (2020) unify synchronous decentralized gradient descent algorithms across various communication topologies, while Even et al. (2024) extend this framework to accommodate asynchronous scenarios. Building on these efforts, Zehtabi et al. (2025) further develops existing frameworks to consider the sporadicity of both communications and local computations. Regarding generalization, Richards et al. (2020) establishes generalization bounds of decentralized SGD in convex settings via uniform stability. Sun et al. (2021) extend this to non-convex settings, revealing an additional $O(\frac{1}{\rho})$ dependence on graph topology, though later empirical studies suggest this gap might be overstated (Kong et al., 2021). To refine this, Zhu et al. (2022) introduce a Gaussian weight difference assumption, improving the $\rho$ dependence to $O((1-\rho)^2)$. Le Bars et al. (2024) further show that in convex settings, the generalization error of local models in decentralized SGD matches that of standard SGD, while in non-convex settings, decentralization primarily affects worst-case generalization. To explain previously unexplained phenomena in decentralized learning (Kong et al., 2021; Gurbuzbalaban et al., 2022; Vogels et al., 2023), Zhu et al. (2023b) later link decentralized SGD to random sharpness-aware minimization, uncovering a flatness bias in decentralized training. Complementing this, Cao et al. (2024) further analyze the flatness properties of DSGD and its role in escaping local minima.

**Decentralized Training of Foundation Models**. DT-FM (Yuan et al., 2022) introduces tasklet scheduling for training Transformers in decentralized settings with low-bandwidth networks, optimizing resource utilization in distributed environments. SWARM Parallelism (Ryabinin et al., 2023) enhances scalability through fault-tolerant pipelines and dynamic node rebalancing. CocktailSGD (Wang et al., 2023a) combines decentralization with sparsification and quantization enhances communication efficiency in fine-tuning LLMs. On the inference side, Petal (Borzunov et al., 2023a) leverages swarm parallelism to amortize inference costs across heterogeneous resources. Recently, Intellect (Jaghouar et al., 2024) built on Diloco (Douillard et al., 2023) has employed a combination of data parallel and model parallel to collaboratively train large models with up to billions of parameters. For a comprehensive overview of large-scale deep learning training, including data, model architecture, optimization strategies, budget constraints, and system design, see Shen et al. (2024).

The following figure presents a comparison between server-based learning and decentralized learning.

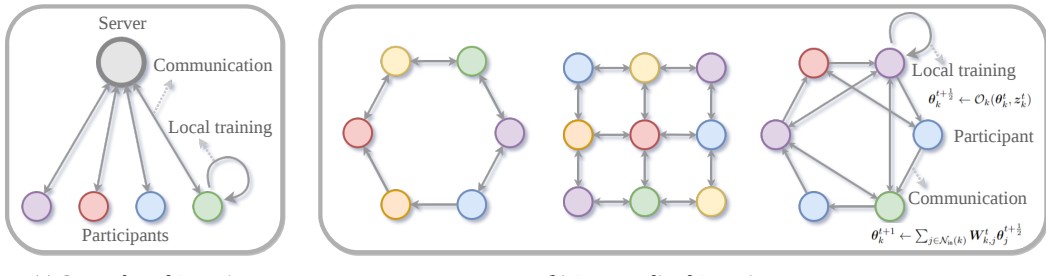

Figure A.1: A comparative illustration of server-based learning versus decentralized learning.

We summarize some commonly used notions regarding decentralized training as follows:

**Definition A.1** (Doubly Stochastic Matrix). Let $\mathcal{G} = (\mathcal{V}, \mathcal{E})$ represent a decentralized communication topology, where $\mathcal{V}$ is the set of $n$ nodes and $\mathcal{E}$ is the set of edges. For any $\mathcal{G} = (\mathcal{V}, \mathcal{E})$, the doubly stochastic gossip matrix $\boldsymbol{W} = [\boldsymbol{W}_{j,k}] \in \mathbb{R}^{n \times n}$ is defined on the edge set $\mathcal{E}$ and satisfies:

- If $j \neq k$ and $(j, k) \notin \mathcal{E}$, then $\boldsymbol{W}_{j,k} = 0$; otherwise, $\boldsymbol{W}_{j,k} > 0$.
- $\boldsymbol{W}_{j,k} \in [0, 1]$ for all $j, k$, and $\sum_k \boldsymbol{W}_{k,j} = \sum_j \boldsymbol{W}_{j,k} = 1$.

Intuitively, the doubly stochastic property ensures a balanced flow of information during gossip communication, a common assumption in decentralized learning literature. However, in the scenarios we consider, participants may occupy different roles within the network. Influential nodes might have higher outgoing weights, i.e., $\sum_{j=1}^{n} \boldsymbol{W}_{j,k} > 1$.

To accommodate such cases while still ensuring the convergence of decentralized SGD (Yuan et al., 2019; Xin et al., 2019), we introduce a relaxed condition:

**Definition A.2** (Row Stochastic Matrix). Let $\mathcal{G} = (\mathcal{V}, \mathcal{E})$ denote a decentralized communication topology, where $\mathcal{V}$ is the set of $n$ nodes and $\mathcal{E}$ is the set of edges. For any $\mathcal{G} = (\mathcal{V}, \mathcal{E})$, the row stochastic gossip matrix $\boldsymbol{W} = [\boldsymbol{W}_{j,k}] \in \mathbb{R}^{n \times n}$ is defined on the edge set $\mathcal{E}$ and satisfies:

- If $j \neq k$ and $(j, k) \notin \mathcal{E}$, then $\boldsymbol{W}_{j,k} = 0$; otherwise, $\boldsymbol{W}_{j,k} > 0$.
- $\boldsymbol{W}_{j,k} \in [0, 1]$ for all $j, k$, and $\sum_j \boldsymbol{W}_{k,j} = 1$.

The weighted adjacency matrix $\boldsymbol{W}$ in Algorithm 1 can vary across iterations, resulting in time-varying collaborations among participants. Additionally, FedAVG (McMahan et al., 2017) is a special case of Algorithm 1 where the averaging step is performed globally. This demonstrates that our framework accommodates decentralized learning with dynamic communication topologies and is applicable to both federated and decentralized learning paradigms, even though the primary focus is on fully decentralized learning without central servers.

### A.3 MULTI-HOP NEIGHBORS

In graph theory, the concept of neighborhoods is fundamental for understanding the structure and dynamics of graphs. To ensure a coherent and comprehensive flow in Section 4, we provide formal definitions of multi-hop neighborhoods.

The adjacency matrix serves as a powerful tool for representing and analyzing the structure of a graph. Multi-hop neighbors can be precisely defined using the adjacency matrix.

**Definition A.3** (Adjacency Matrix). The adjacency matrix $A$ of a graph $G = (\mathcal{V}, \mathcal{E})$ is an $n \times n$ square matrix (where $n = |\mathcal{V}|$) defined by:

$$A_{jk} = \begin{cases} 1 & \text{if } (j, k) \in \mathcal{E}, \\ 0 & \text{otherwise.} \end{cases} \tag{A.1}$$

The adjacency matrix enables the determination of $r$-hop neighbors through matrix exponentiation. Specifically, the $(j, k)$-entry of $A^r$, denoted as $(A^r)_{jk}$, corresponds to the number of distinct paths of length $r$ from node $j$ to node $k$.

**Definition A.4** ($r$-hop Neighbor via Adjacency Matrix). The set of $r$-hop neighbors is formally defined using the adjacency matrix $A$ as:

$$\mathcal{N}^{(r)}(j) = \left\{ k \in \mathcal{V} \,\middle|\, (A^r)_{jk} > 0 \text{ and } \forall s < r, \, (A^s)_{jk} = 0 \right\}. \tag{A.2}$$

This definition indicates that there exists at least one path of length $r$ connecting nodes $j$ and $k$, and no shorter path exists between them.

Multi-hop neighbors can also be defined via the *shortest path length* between two nodes.

**Definition A.5** (Shortest Path Length). In a connected graph $G = (\mathcal{V}, \mathcal{E})$, the shortest path length $d(j, k)$ between nodes $j \in \mathcal{V}$ and $k \in \mathcal{V}$ is the minimum number of edges that must be traversed to travel from $j$ to $k$.

Building upon this, the set of $r$-hop neighbors is defined as follows:

**Definition A.6** ($r$-hop Neighbor via Shortest Path Length). For any node $j \in \mathcal{V}$ and a positive integer $r \geq 1$, the set of $r$-hop neighbors, denoted by $\mathcal{N}^{(r)}(j)$, consists of all nodes that are at a distance of exactly $r$ from node $j$. Formally,

$$\mathcal{N}^{(r)}(j) = \{k \in \mathcal{V} \mid d(j, k) = r\}, \tag{A.3}$$

where $d(j, k)$ represents the shortest path length between nodes $j$ and $k$ in the graph $G$.

Furthermore, an alternative perspective on $r$-hop neighborhoods involves characterizing them through sequences of nodes, which provides a formal framework aligned with influence propagation in decentralized learning.

**Definition A.7** ($r$-hop Neighbor via Node Sequences). For any node $j \in \mathcal{V}$ and a positive integer $r \geq 1$, let $P_j^{(r)}$ denote the set of all sequences $(k_1, \ldots, k_r)$ such that for each $s = 1, \ldots, r$, the node $k_s$ is an out-neighbor of $k_{s-1}$, with $k_0 = j$. Formally,

$$P_j^{(r)} = \left\{ (k_1, \ldots, k_r) \mid k_s \in \mathcal{N}_{\text{out}}^{(1)}(k_{s-1}) \text{ for } s = 1, \ldots, r \right\}.$$

This definition ensures that each node in the $r$-hop neighborhood is reachable from node $j$ through a sequence of consecutive immediate out-neighbors within $\rho \leq r$ steps.

This sequence-based characterization of $r$-hop neighborhoods provides a granular understanding of the pathways through which influence or information can propagate within the network, complementing the previous definitions based on adjacency matrices and shortest path lengths.

# B  DISCUSSIONS

## B.1  PRACTICAL APPLICATIONS OF DICE

**Decentralized Machine Unlearning**. As concerns about data privacy and the right to be forgotten increase, the ability to remove specific data contributions from a trained model becomes important (Guo et al., 2020; Sekhari et al., 2021). In decentralized settings, retraining the model from scratch is often impractical for edge users with limited compute. The proximal influence measure enables participants to estimate the impact of removing a particular data instance from its neighbor. For example, by assessing the influence of $z_j^t$ on neighbors, participants can adjust their local models to mitigate the effects of $z_j^t$ without requesting full retraining of the whole decentralized learning system. This approach facilitates efficient and targeted unlearning procedures, avoiding costly system-wide retraining while respecting individual data privacy requests.

### B.2 ADDITIONAL RELATED WORK

**Clustered Federated Learning**. Clustered Federated Learning (CFL) addresses the challenge of data heterogeneity by grouping clients with similar data distributions and training separate models for each cluster (Mansour et al., 2020; Ghosh et al., 2020; Sattler et al., 2021; Kim et al., 2024). Gradient-based CFL methods (Sattler et al., 2021; Kim et al., 2024) use client gradient similarities to form clusters, with Sattler et al. (2021) employing cosine similarity to recursively partition clients after convergence and Kim et al. (2024) dynamically applying spectral clustering to organize clients based on gradient features during training. These methods effectively capture direct, peer-to-peer gradient relationships to cluster clients with similar data-generating distributions. Both gradient-based CFL and the one-hop DICE estimator (see Proposition 1) utilize gradient similarity information. However, CFL is inherently limited to local interactions, as its gradient similarity metrics are confined to pairwise relationships. In contrast, DICE extends far beyond this scope by quantifying the propagation of influence across multiple hops in a decentralized network. Mathematically, Theorem 2 highlights how DICE generalizes peer-level gradient similarity into a non-trivial extension for decentralized networks. This includes incorporating key factors including network topology and curvature information, enabling a deeper understanding of how influence flows through the whole decentralized learning systems. A promising future direction is to explore the potential of DICE-E as a more advanced high-order gradient similarity metric for effectively clustering participants in decentralized federated learning.

## C PROOF

### C.1 PROOF OF PROPOSITION 1

**Proposition 1** (Approximation of One-hop DICE-GT). The one-hop DICE-GT value (see Definition 2) can be linearly approximated as follows:

$$\mathcal{I}_{\text{DICE-E}}^{(1)}(\boldsymbol{z}_j^t, \boldsymbol{z}') = -q_j \nabla L(\boldsymbol{\theta}_j^t; \boldsymbol{z}')^\top \Delta_j(\boldsymbol{\theta}_j^t, \boldsymbol{z}_j^t) - \sum_{k \in \mathcal{N}_{\text{out}}^{(1)}(j)} q_k \, \boldsymbol{W}_{k,j}^t \, \nabla L(\boldsymbol{\theta}_k^{t+1}; \boldsymbol{z}')^\top \Delta_j(\boldsymbol{\theta}_j^t, \boldsymbol{z}_j^t),$$

(C.1)

where $\Delta_j(\boldsymbol{\theta}_j^t, \boldsymbol{z}_j^t) = \mathcal{O}_j(\boldsymbol{\theta}_j^t, \boldsymbol{z}_j^t) - \boldsymbol{\theta}_j^t$. The proof is given below.

*Proof.* Recall from Definition 2 that the one-hop DICE-GT is defined by

$$\mathcal{I}_{\text{DICE-GT}}^{(1)}(\boldsymbol{z}_j^t, \boldsymbol{z}') = q_j \left( L(\boldsymbol{\theta}_j^{t+\frac{1}{2}}; \boldsymbol{z}') - L(\boldsymbol{\theta}_j^t; \boldsymbol{z}') \right) + \sum_{k \in \mathcal{N}_{\text{out}}^{(1)}(j)} q_k \left( L(\boldsymbol{\theta}_k^{t+1}; \boldsymbol{z}') - L(\boldsymbol{\theta}_{k \setminus \boldsymbol{z}_j^t}^{t+1}; \boldsymbol{z}') \right).$$

We proceed by applying a first-order Taylor expansion to each term.

**First term:** Using Taylor expansion, we write

$$L(\boldsymbol{\theta}_j^{t+\frac{1}{2}}; \boldsymbol{z}') - L(\boldsymbol{\theta}_j^t; \boldsymbol{z}') \approx \nabla L(\boldsymbol{\theta}_j^t; \boldsymbol{z}')^\top \left( \boldsymbol{\theta}_j^{t+\frac{1}{2}} - \boldsymbol{\theta}_j^t \right).$$

Under the new update rule, we have

$$\boldsymbol{\theta}_j^{t+\frac{1}{2}} = \mathcal{O}_j(\boldsymbol{\theta}_j^t, \boldsymbol{z}_j^t),$$

so that

$$\boldsymbol{\theta}_j^{t+\frac{1}{2}} - \boldsymbol{\theta}_j^t = \mathcal{O}_j(\boldsymbol{\theta}_j^t, \boldsymbol{z}_j^t) - \boldsymbol{\theta}_j^t.$$

Thus, the first term is approximated by

$$L(\boldsymbol{\theta}_j^{t+\frac{1}{2}}; \boldsymbol{z}') - L(\boldsymbol{\theta}_j^t; \boldsymbol{z}') \approx \nabla L(\boldsymbol{\theta}_j^t; \boldsymbol{z}')^\top \left( \mathcal{O}_j(\boldsymbol{\theta}_j^t, \boldsymbol{z}_j^t) - \boldsymbol{\theta}_j^t \right).$$

**Second term:** For each $k \in \mathcal{N}_{\text{out}}^{(1)}(j)$, we similarly have

$$L(\boldsymbol{\theta}_k^{t+1}; \boldsymbol{z}') - L(\boldsymbol{\theta}_{k \setminus \boldsymbol{z}_j^t}^{t+1}; \boldsymbol{z}') \approx \nabla L(\boldsymbol{\theta}_k^{t+1}; \boldsymbol{z}')^\top \left( \boldsymbol{\theta}_k^{t+1} - \boldsymbol{\theta}_{k \setminus \boldsymbol{z}_j^t}^{t+1} \right).$$

By the gossip averaging step in the algorithm, we have

$$\boldsymbol{\theta}_k^{t+1} = \sum_{l \in \mathcal{N}_{\text{in}}(k)} \boldsymbol{W}_{k,l}^t \, \boldsymbol{\theta}_l^{t+\frac{1}{2}},$$

$$\boldsymbol{\theta}_{k \setminus \boldsymbol{z}_j^t}^{t+1} = \boldsymbol{W}_{k,j}^t \, \boldsymbol{\theta}_j^t + \sum_{l \in \mathcal{N}_{\text{in}}(k) \setminus \{j\}} \boldsymbol{W}_{k,l}^t \, \boldsymbol{\theta}_l^{t+\frac{1}{2}}.$$

It then follows that

$$\boldsymbol{\theta}_k^{t+1} - \boldsymbol{\theta}_{k \setminus \boldsymbol{z}_j^t}^{t+1} = \boldsymbol{W}_{k,j}^t \left( \boldsymbol{\theta}_j^{t+\frac{1}{2}} - \boldsymbol{\theta}_j^t \right) = \boldsymbol{W}_{k,j}^t \left( \mathcal{O}_j(\boldsymbol{\theta}_j^t, \boldsymbol{z}_j^t) - \boldsymbol{\theta}_j^t \right).$$

Thus, the second term becomes

$$L(\boldsymbol{\theta}_k^{t+1}; \boldsymbol{z}') - L(\boldsymbol{\theta}_{k \setminus \boldsymbol{z}_j^t}^{t+1}; \boldsymbol{z}') \approx \boldsymbol{W}_{k,j}^t \, \nabla L(\boldsymbol{\theta}_k^{t+1}; \boldsymbol{z}')^\top \left( \mathcal{O}_j(\boldsymbol{\theta}_j^t, \boldsymbol{z}_j^t) - \boldsymbol{\theta}_j^t \right).$$

**Combining the Approximations:** Substituting the above approximations into the definition of $\mathcal{I}_{\text{DICE-GT}}^{(1)}$ yields

$$\mathcal{I}_{\text{DICE-E}}^{(1)}(\boldsymbol{z}_j^t, \boldsymbol{z}') \approx -q_j \, \nabla L(\boldsymbol{\theta}_j^t; \boldsymbol{z}')^\top \left( \mathcal{O}_j(\boldsymbol{\theta}_j^t, \boldsymbol{z}_j^t) - \boldsymbol{\theta}_j^t \right)$$

$$- \sum_{k \in \mathcal{N}_{\text{out}}^{(1)}(j)} q_k \, \boldsymbol{W}_{k,j}^t \, \nabla L(\boldsymbol{\theta}_k^{t+1}; \boldsymbol{z}')^\top \left( \mathcal{O}_j(\boldsymbol{\theta}_j^t, \boldsymbol{z}_j^t) - \boldsymbol{\theta}_j^t \right).$$

This completes the proof. $\qquad \square$

## C.2 Proof of Two-hop DICE-E Approximation

**Proposition 2** (Approximation of Two-hop DICE-GT)**.** The two-hop DICE-GT influence $\mathcal{I}_{\text{DICE-E}}^{(2)}(\boldsymbol{z}_j^t, \boldsymbol{z}')$ (see Definition 3) can be approximated as

$$\mathcal{I}_{\text{DICE-E}}^{(2)}(\boldsymbol{z}_j^t, \boldsymbol{z}') = \mathcal{I}_{\text{DICE-E}}^{(1)}(\boldsymbol{z}_j^t, \boldsymbol{z}')$$
$$- \sum_{k \in \mathcal{N}_{\text{out}}^{(1)}(j)} \sum_{l \in \mathcal{N}_{\text{out}}^{(1)}(k)} \eta^t q_l \boldsymbol{W}_{l,k}^{t+1} \boldsymbol{W}_{k,j}^t \nabla L(\boldsymbol{\theta}_l^{t+2}; \boldsymbol{z}')^\top (\boldsymbol{I} - \eta^{t+1} \boldsymbol{H}(\boldsymbol{\theta}_k^{t+1}; \boldsymbol{z}_k^{t+1})) \Delta_j(\boldsymbol{\theta}_j^t; \boldsymbol{z}_j^t),$$

$$(\text{C.2})$$

where $\Delta_j(\boldsymbol{\theta}_j^t, \boldsymbol{z}_j^t) \triangleq \mathcal{O}_j(\boldsymbol{\theta}_j^t, \boldsymbol{z}_j^t) - \boldsymbol{\theta}_j^t$. and $\boldsymbol{H}(\boldsymbol{\theta}_k^{t+1}; \boldsymbol{z}_k^{t+1})$ denotes the Hessian matrix of $L$ with respect to $\boldsymbol{\theta}_k^{t+1}$ evaluated at $\boldsymbol{z}_k^{t+1}$.

*Proof.* We begin from the definition in Definition 3 where the two-hop DICE-GT influence is given by

$$\mathcal{I}_{\text{DICE-GT}}^{(2)}(\boldsymbol{z}_j^t, \boldsymbol{z}') = \sum_{k \in \mathcal{N}_{\text{out}}^{(1)}(j)} \sum_{l \in \mathcal{N}_{\text{out}}^{(1)}(k)} q_l \left[ L(\boldsymbol{\theta}_l^{t+2}; \boldsymbol{z}') - L(\boldsymbol{\theta}_{l \setminus \boldsymbol{z}_j^t}^{t+2}; \boldsymbol{z}') \right].$$

Subtracting the one-hop influence from both sides yields

$$\mathcal{I}_{\text{DICE-GT}}^{(2)}(\boldsymbol{z}_j^t, \boldsymbol{z}') - \mathcal{I}_{\text{DICE-GT}}^{(1)}(\boldsymbol{z}_j^t, \boldsymbol{z}') = \sum_{k \in \mathcal{N}_{\text{out}}^{(1)}(j)} \sum_{l \in \mathcal{N}_{\text{out}}^{(1)}(k)} q_l \left[ L(\boldsymbol{\theta}_l^{t+2}; \boldsymbol{z}') - L(\boldsymbol{\theta}_{l \setminus \boldsymbol{z}_j^t}^{t+2}; \boldsymbol{z}') \right].$$

A first-order Taylor expansion gives

$$L(\boldsymbol{\theta}_l^{t+2}; \boldsymbol{z}') - L(\boldsymbol{\theta}_{l\backslash \boldsymbol{z}_j^t}^{t+2}; \boldsymbol{z}') \approx \nabla L(\boldsymbol{\theta}_l^{t+2}; \boldsymbol{z}')^\top \left(\boldsymbol{\theta}_l^{t+2} - \boldsymbol{\theta}_{l\backslash \boldsymbol{z}_j^t}^{t+2}\right).$$

Next, the update rule in Algorithm 1 implies that

$$\boldsymbol{\theta}_l^{t+2} = \sum_{m \in \mathcal{N}_{\mathrm{in}}(l)} \boldsymbol{W}_{l,m}^{t+1} \boldsymbol{\theta}_m^{t+\frac{3}{2}},$$

and similarly for $\boldsymbol{\theta}_{l\backslash \boldsymbol{z}_j^t}^{t+2}$. Since the influence of $\boldsymbol{z}_j^t$ reaches $l$ only via intermediate nodes, we may write

$$\boldsymbol{\theta}_l^{t+2} - \boldsymbol{\theta}_{l\backslash \boldsymbol{z}_j^t}^{t+2} = \sum_{k \in \mathcal{N}_{\mathrm{out}}^{(1)}(j)} \boldsymbol{W}_{l,k}^{t+1} \left(\boldsymbol{\theta}_k^{t+\frac{3}{2}} - \boldsymbol{\theta}_{k\backslash \boldsymbol{z}_j^t}^{t+\frac{3}{2}}\right).$$

At an intermediate node $k$, the local update in iteration $t+1$ gives

$$\boldsymbol{\theta}_k^{t+\frac{3}{2}} = \boldsymbol{\theta}_k^{t+1} - \eta^{t+1} \nabla L(\boldsymbol{\theta}_k^{t+1}; \boldsymbol{z}_k^{t+1}).$$

Therefore, the difference between the actual and perturbed updates is

$$\boldsymbol{\theta}_k^{t+\frac{3}{2}} - \boldsymbol{\theta}_{k\backslash \boldsymbol{z}_j^t}^{t+\frac{3}{2}} \approx \left(\boldsymbol{I} - \eta^{t+1} \boldsymbol{H}(\boldsymbol{\theta}_k^{t+1}; \boldsymbol{z}_k^{t+1})\right) \left(\boldsymbol{\theta}_k^{t+1} - \boldsymbol{\theta}_{k\backslash \boldsymbol{z}_j^t}^{t+1}\right).$$

At node $k$, the difference between the parameters updated with and without the influence from $z_j^t$ is given by

$$\boldsymbol{\theta}_k^{t+1} - \boldsymbol{\theta}_{k\backslash \boldsymbol{z}_j^t}^{t+1} = \boldsymbol{W}_{k,j}^t \left(\boldsymbol{\theta}_j^{t+\frac{1}{2}} - \boldsymbol{\theta}_j^t\right) \approx -\eta^t \boldsymbol{W}_{k,j}^t \Delta_j(\boldsymbol{\theta}_j^t, \boldsymbol{z}_j^t).$$

Hence, we obtain

$$\boldsymbol{\theta}_k^{t+\frac{3}{2}} - \boldsymbol{\theta}_{k\backslash \boldsymbol{z}_j^t}^{t+\frac{3}{2}} \approx -\eta^t \left(\boldsymbol{I} - \eta^{t+1} \boldsymbol{H}(\boldsymbol{\theta}_k^{t+1}; \boldsymbol{z}_k^{t+1})\right) \boldsymbol{W}_{k,j}^t \Delta_j(\boldsymbol{\theta}_j^t, \boldsymbol{z}_j^t).$$

Substituting back into the expression for node $l$, we have

$$\boldsymbol{\theta}_l^{t+2} - \boldsymbol{\theta}_{l\backslash \boldsymbol{z}_j^t}^{t+2} \approx -\eta^t \sum_{k \in \mathcal{N}_{\mathrm{out}}^{(1)}(j)} \boldsymbol{W}_{l,k}^{t+1} \boldsymbol{W}_{k,j}^t \left(\boldsymbol{I} - \eta^{t+1} \boldsymbol{H}(\boldsymbol{\theta}_k^{t+1}; \boldsymbol{z}_k^{t+1})\right) \Delta_j(\boldsymbol{\theta}_j^t, \boldsymbol{z}_j^t).$$

Plugging this into the Taylor expansion for the loss difference and multiplying by $q_l$ yields

$$L(\boldsymbol{\theta}_l^{t+2}; \boldsymbol{z}') - L(\boldsymbol{\theta}_{l\backslash \boldsymbol{z}_j^t}^{t+2}; \boldsymbol{z}')$$
$$\approx -\eta^t \nabla L(\boldsymbol{\theta}_l^{t+2}; \boldsymbol{z}')^\top \sum_{k \in \mathcal{N}_{\mathrm{out}}^{(1)}(j)} \boldsymbol{W}_{l,k}^{t+1} \boldsymbol{W}_{k,j}^t \left(\boldsymbol{I} - \eta^{t+1} \boldsymbol{H}(\boldsymbol{\theta}_k^{t+1}; \boldsymbol{z}_k^{t+1})\right) \Delta_j(\boldsymbol{\theta}_j^t, \boldsymbol{z}_j^t).$$

Finally, summing over all intermediate nodes and multiplying by $q_l$, we obtain

$$\mathcal{I}_{\mathrm{DICE\text{-}E}}^{(2)}(\boldsymbol{z}_j^t, \boldsymbol{z}') - \mathcal{I}_{\mathrm{DICE\text{-}E}}^{(1)}(\boldsymbol{z}_j^t, \boldsymbol{z}')$$
$$\approx - \sum_{k \in \mathcal{N}_{\mathrm{out}}^{(1)}(j)} \sum_{l \in \mathcal{N}_{\mathrm{out}}^{(1)}(k)} \eta^t q_l \boldsymbol{W}_{l,k}^{t+1} \boldsymbol{W}_{k,j}^t \nabla L(\boldsymbol{\theta}_l^{t+2}; \boldsymbol{z}')^\top \left(\boldsymbol{I} - \eta^{t+1} \boldsymbol{H}(\boldsymbol{\theta}_k^{t+1}; \boldsymbol{z}_k^{t+1})\right) \Delta_j(\boldsymbol{\theta}_j^t, \boldsymbol{z}_j^t).$$

This completes the proof. $\qquad\square$

## C.3 PROOF OF THEOREM 2

**Theorem 2** (Approximation of Multi-hop DICE-GT). The $r$-hop DICE-GT value (see Definition 3) can be approximated as

$$
\mathcal{I}_{\text{DICE-E}}^{(r)}(\boldsymbol{z}_j^t, \boldsymbol{z}') = -\sum_{\rho=0}^{r} \sum_{(k_1,\ldots,k_\rho)\in P_j^{(\rho)}} \eta^t\, q_{k_\rho} \left(\prod_{s=1}^{\rho} \boldsymbol{W}_{k_s,k_{s-1}}^{t+s-1}\right) \nabla L\big(\boldsymbol{\theta}_{k_\rho}^{t+\rho}; \boldsymbol{z}'\big)^\top
$$
$$
\times \left(\prod_{s=2}^{\rho}\Big(\boldsymbol{I} - \eta^{t+s-1}\,\boldsymbol{H}\big(\boldsymbol{\theta}_{k_s}^{t+s-1}; \boldsymbol{z}_{k_s}^{t+s-1}\big)\Big)\right) \Delta_j(\boldsymbol{\theta}_j^t, \boldsymbol{z}_j^t), \tag{C.3}
$$

where

$$
\Delta_j(\boldsymbol{\theta}_j^t, \boldsymbol{z}_j^t) \triangleq \mathcal{O}_j(\boldsymbol{\theta}_j^t, \boldsymbol{z}_j^t) - \boldsymbol{\theta}_j^t,
$$

where $k_0 = j$, $P_j^{(\rho)}$ denotes the set of all sequences $(k_1,\ldots,k_\rho)$ such that $k_s \in \mathcal{N}_{\text{out}}^{(1)}(k_{s-1})$ for $s = 1,\ldots,\rho$ (see Definition A.7) and $\boldsymbol{H}(\boldsymbol{\theta}_{k_s}^{t+s}; \boldsymbol{z}_{k_s}^{t+s})$ is the Hessian matrix of $L$ with respect to $\boldsymbol{\theta}$ evaluated at $\boldsymbol{\theta}_{k_s}^{t+s}$ and data $\boldsymbol{z}_{k_s}^{t+s}$. For the cases when $\rho = 0$ and $\rho = 1$, the relevant product expressions are defined as identity matrices, thereby ensuring that the r-hop DICE-E remains well-defined.

*Proof.* From the definition in Definition 3, the $r$-hop influence is

$$
\mathcal{I}_{\text{DICE-GT}}^{(r)}(\boldsymbol{z}_j^t, \boldsymbol{z}') = \sum_{\rho=0}^{r} \sum_{(k_1,\ldots,k_\rho)\in P_j^{(\rho)}} q_{k_\rho}\Big(L(\boldsymbol{\theta}_{k_\rho}^{t+\rho}; \boldsymbol{z}') - L(\boldsymbol{\theta}_{k_\rho\backslash\boldsymbol{z}_j^t}^{t+\rho}; \boldsymbol{z}')\Big).
$$

Here the $\rho = 0$ term (with $k_0 = j$) corresponds to the direct influence on node $j$. For any $\rho \geq 1$, define the incremental influence as

$$
\Delta\mathcal{I}_{\text{DICE-GT}}^{(\rho)}(\boldsymbol{z}_j^t, \boldsymbol{z}') = \mathcal{I}_{\text{DICE-GT}}^{(\rho)}(\boldsymbol{z}_j^t, \boldsymbol{z}') - \mathcal{I}_{\text{DICE-GT}}^{(\rho-1)}(\boldsymbol{z}_j^t, \boldsymbol{z}').
$$

Thus,

$$
\Delta\mathcal{I}_{\text{DICE-GT}}^{(\rho)}(\boldsymbol{z}_j^t, \boldsymbol{z}') = \sum_{(k_1,\ldots,k_\rho)\in P_j^{(\rho)}} q_{k_\rho}\left[L(\boldsymbol{\theta}_{k_\rho}^{t+\rho}; \boldsymbol{z}') - L(\boldsymbol{\theta}_{k_\rho\backslash\boldsymbol{z}_j^t}^{t+\rho}; \boldsymbol{z}')\right].
$$

A first-order Taylor expansion gives

$$
L(\boldsymbol{\theta}_{k_\rho}^{t+\rho}; \boldsymbol{z}') - L(\boldsymbol{\theta}_{k_\rho\backslash\boldsymbol{z}_j^t}^{t+\rho}; \boldsymbol{z}') \approx \nabla L\big(\boldsymbol{\theta}_{k_\rho}^{t+\rho}; \boldsymbol{z}'\big)^\top \left[\boldsymbol{\theta}_{k_\rho}^{t+\rho} - \boldsymbol{\theta}_{k_\rho\backslash\boldsymbol{z}_j^t}^{t+\rho}\right].
$$

Our goal is to express the parameter change $\Delta\boldsymbol{\theta}_{k_\rho} \triangleq \boldsymbol{\theta}_{k_\rho}^{t+\rho} - \boldsymbol{\theta}_{k_\rho\backslash\boldsymbol{z}_j^t}^{t+\rho}$ in terms of the propagated perturbation from node $j$.

According to the gossip update in Algorithm 1, for any node $k_\rho$ we have

$$
\boldsymbol{\theta}_{k_\rho}^{t+\rho} = \sum_{m\in\mathcal{N}_{\text{in}}(k_\rho)} \boldsymbol{W}_{k_\rho,m}^{t+\rho-1}\,\boldsymbol{\theta}_m^{t+\rho-\frac{1}{2}},
$$
$$
\boldsymbol{\theta}_{k_\rho\backslash\boldsymbol{z}_j^t}^{t+\rho} = \sum_{m\in\mathcal{N}_{\text{in}}(k_\rho)} \boldsymbol{W}_{k_\rho,m}^{t+\rho-1}\,\boldsymbol{\theta}_{m\backslash\boldsymbol{z}_j^t}^{t+\rho-\frac{1}{2}}.
$$

Since only the predecessor $k_{\rho-1}$ is affected by the perturbation from $\boldsymbol{z}_j^t$, we obtain

$$
\Delta\boldsymbol{\theta}_{k_\rho} = \boldsymbol{W}_{k_\rho,k_{\rho-1}}^{t+\rho-1}\left(\boldsymbol{\theta}_{k_{\rho-1}}^{t+\rho-\frac{1}{2}} - \boldsymbol{\theta}_{k_{\rho-1}\backslash\boldsymbol{z}_j^t}^{t+\rho-\frac{1}{2}}\right).
$$

At node $k_{\rho-1}$, using the local update rule,

$$
\boldsymbol{\theta}_{k_{\rho-1}}^{t+\rho-\frac{1}{2}} = \mathcal{O}_{k_{\rho-1}}(\boldsymbol{\theta}_{k_{\rho-1}}^{t+\rho-1}, \boldsymbol{z}_{k_{\rho-1}}^{t+\rho-1}),
$$

the difference can be written as

$$\boldsymbol{\theta}_{k_{\rho-1}}^{t+\rho-\frac{1}{2}} - \boldsymbol{\theta}_{k_{\rho-1}\setminus\boldsymbol{z}_j^t}^{t+\rho-\frac{1}{2}} = \left(\boldsymbol{\theta}_{k_{\rho-1}}^{t+\rho-1} - \boldsymbol{\theta}_{k_{\rho-1}\setminus\boldsymbol{z}_j^t}^{t+\rho-1}\right)$$
$$- \eta^{t+\rho-1}\left(\nabla L(\boldsymbol{\theta}_{k_{\rho-1}}^{t+\rho-1}; \boldsymbol{z}_{k_{\rho-1}}^{t+\rho-1}) - \nabla L(\boldsymbol{\theta}_{k_{\rho-1}\setminus\boldsymbol{z}_j^t}^{t+\rho-1}; \boldsymbol{z}_{k_{\rho-1}}^{t+\rho-1})\right).$$

A further first-order Taylor expansion approximates

$$\nabla L(\boldsymbol{\theta}_{k_{\rho-1}}^{t+\rho-1}; \boldsymbol{z}_{k_{\rho-1}}^{t+\rho-1}) - \nabla L(\boldsymbol{\theta}_{k_{\rho-1}\setminus\boldsymbol{z}_j^t}^{t+\rho-1}; \boldsymbol{z}_{k_{\rho-1}}^{t+\rho-1}) \approx \boldsymbol{H}_{k_{\rho-1}}^{t+\rho-1}\left(\boldsymbol{\theta}_{k_{\rho-1}}^{t+\rho-1} - \boldsymbol{\theta}_{k_{\rho-1}\setminus\boldsymbol{z}_j^t}^{t+\rho-1}\right).$$

Thus,

$$\Delta\boldsymbol{\theta}_{k_\rho} \approx \boldsymbol{W}_{k_\rho,k_{\rho-1}}^{t+\rho-1}\left(\boldsymbol{I} - \eta^{t+\rho-1}\boldsymbol{H}_{k_{\rho-1}}^{t+\rho-1}\right)\left(\boldsymbol{\theta}_{k_{\rho-1}}^{t+\rho-1} - \boldsymbol{\theta}_{k_{\rho-1}\setminus\boldsymbol{z}_j^t}^{t+\rho-1}\right).$$

By recursively unrolling this relation from $s = \rho$ down to $s = 1$, we deduce

$$\boldsymbol{\theta}_{k_\rho}^{t+\rho} - \boldsymbol{\theta}_{k_\rho\setminus\boldsymbol{z}_j^t}^{t+\rho} \approx \left(\prod_{s=1}^{\rho}\boldsymbol{W}_{k_s,k_{s-1}}^{t+s-1}\prod_{s=2}^{\rho}\left(\boldsymbol{I} - \eta^{t+s-1}\boldsymbol{H}_{k_{s-1}}^{t+s-1}\right)\right)\left(\boldsymbol{\theta}_{k_1}^{t+1} - \boldsymbol{\theta}_{k_1\setminus\boldsymbol{z}_j^t}^{t+1}\right).$$

At the base level, the local update at node $j$ gives

$$\boldsymbol{\theta}_{k_1}^{t+1} - \boldsymbol{\theta}_{k_1\setminus\boldsymbol{z}_j^t}^{t+1} = -\eta^t\,\boldsymbol{W}_{k_1,j}^t\,\Delta_j(\boldsymbol{\theta}_j^t, \boldsymbol{z}_j^t).$$

Hence,

$$\boldsymbol{\theta}_{k_\rho}^{t+\rho} - \boldsymbol{\theta}_{k_\rho\setminus\boldsymbol{z}_j^t}^{t+\rho} \approx -\eta^t\left(\prod_{s=1}^{\rho}\boldsymbol{W}_{k_s,k_{s-1}}^{t+s-1}\right)\left(\prod_{s=2}^{\rho}\left(\boldsymbol{I} - \eta^{t+s-1}\boldsymbol{H}_{k_{s-1}}^{t+s-1}\right)\right)\Delta_j(\boldsymbol{\theta}_j^t, \boldsymbol{z}_j^t).$$

Substituting this back into the Taylor expansion for the loss difference, we have

$$\Delta\mathcal{I}_{\text{DICE-GT}}^{(\rho)}(\boldsymbol{z}_j^t, \boldsymbol{z}') \approx -\sum_{(k_1,\ldots,k_\rho)\in P_j^{(\rho)}}\eta^t\,q_{k_\rho}\left(\prod_{s=1}^{\rho}\boldsymbol{W}_{k_s,k_{s-1}}^{t+s-1}\right)\nabla L(\boldsymbol{\theta}_{k_\rho}^{t+\rho}; \boldsymbol{z}')^\top$$
$$\times\left(\prod_{s=2}^{\rho}\left(\boldsymbol{I} - \eta^{t+s-1}\boldsymbol{H}_{k_{s-1}}^{t+s-1}\right)\right)\Delta_j(\boldsymbol{\theta}_j^t, \boldsymbol{z}_j^t).$$

Summing over $\rho = 0$ to $r$ (with the $\rho = 0$ term accounting for the direct influence at node $j$) yields

$$\mathcal{I}_{\text{DICE-E}}^{(r)}(\boldsymbol{z}_j^t, \boldsymbol{z}') = -\sum_{\rho=0}^{r}\sum_{(k_1,\ldots,k_\rho)\in P_j^{(\rho)}}\eta^t\,q_{k_\rho}\left(\prod_{s=1}^{\rho}\boldsymbol{W}_{k_s,k_{s-1}}^{t+s-1}\right)\nabla L(\boldsymbol{\theta}_{k_\rho}^{t+\rho}; \boldsymbol{z}')^\top$$
$$\times\left(\prod_{s=2}^{\rho}\left(\boldsymbol{I} - \eta^{t+s-1}\boldsymbol{H}(\boldsymbol{\theta}_{k_s}^{t+s-1}; \boldsymbol{z}_{k_s}^{t+s-1})\right)\right)\Delta_j(\boldsymbol{\theta}_j^t, \boldsymbol{z}_j^t).$$

This concludes the proof. $\qquad\square$

## D  ADDITIONAL EXPERIMENTS

### D.1  DETAILS OF EXPERIMENTAL SETUP

**Computational Resources.** The experiments are conducted on a computing facility equipped with 80 GB NVIDIA® A100™ GPUs.

We employ the vanilla mini-batch Adapt-Then-Communicate version of Decentralized SGD ((Lopes & Sayed, 2008), see Algorithm 1) with commonly used network topologies (Ying et al., 2021) to train

three-layer MLPs (Rumelhart et al., 1986), three-layer CNNs (LeCun et al., 1998), and ResNet-18 (He et al., 2016) on subsets of MNIST (LeCun et al., 1998), CIFAR-10, CIFAR-100 (Krizhevsky et al., 2009), and Tiny ImageNet (Le & Yang, 2015). The number of participants (one GPU as a participant) is set to 16 and 32, with each participant holding 512 samples. For sensitivity analysis, we evaluate the stability of results under hyperparameter adjustments. The local batch size is varied as 16, 64, and 128 per participant, while the learning rate is set as 0.1 and 0.01 without decay. The code will be made publicly available.

## D.2 INFLUENCE ALIGNMENT

In this experiments, we evaluate the alignment between one-hop DICE-GT (see Definition 2) and its first-order approximation, one-hop DICE-E (see Proposition 1). One-hop DICE-E $\mathcal{I}^{(1)}_{\text{DICE-E}}(\mathcal{B}^t_j, \boldsymbol{z}')$ is computed as the sum of one-sample DICE-E within the mini-batch $\mathcal{B}^t_j$ thanks to the additivity (see Equation (4)). DICE-GT $\mathcal{I}_{\text{DICE-GT}^{(1)}}(\mathcal{B}^t_j, \boldsymbol{z}')$ is calculated by measuring the loss reduction after removing $\mathcal{B}^t_j$ from node $j$ at the $t$-th iteration. In the following Figures, each plot contains 30 points, with each point representing the result of a single comparison of one-hop DICE-GT and the estimated influence DICE-E. Strong alignments of DICE-GT and DICE-E are observed across datasets (CIFAR-10, CIFAR-100 and Tiny ImageNet) and model architectures (CNN and MLP).

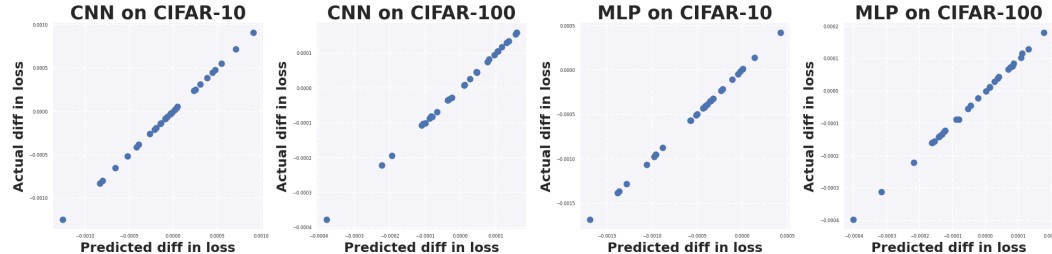

Figure D.1: Alignment between one-hop DICE-GT (vertical axis) and DICE-E (horizontal axis) on a 16-node exponential graph. Each node uses a 512-sample subset of CIFAR-10 or CIFAR-100. Models are trained for 5 epochs with a batch size of 128 and a learning rate of 0.1.

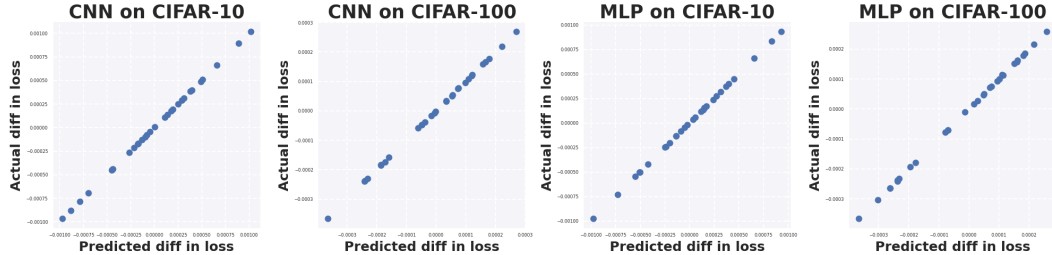

Figure D.2: Alignment between one-hop DICE-GT (vertical axis) and DICE-E (horizontal axis) on a 32-node exponential graph. Each node uses a 512-sample subset of CIFAR-10 or CIFAR-100. Models are trained for 5 epochs with a batch size of 128 and a learning rate of 0.1.

We conduct additional sensitivity analysis experiments to evaluate the robustness of DICE-E under varying hyperparameters, including learning rate, batch size, and training epoch. These results demonstrate that DICE-E provides a strong approximation of DICE-GT, achieving consistent alignment across datasets (CIFAR-10 and CIFAR-100) and model architectures (CNN and MLP) under different batch sizes, learning rates, and training epochs.

### D.2.1 SENSITIVITY ANALYSIS ON BATCH SIZE

We conduct experiments to evaluate the robustness of DICE-E under varying batch sizes.

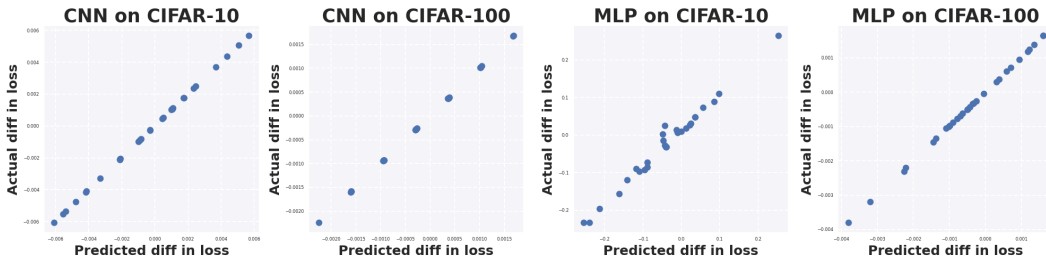

Figure D.3: Alignment between one-hop DICE-GT (vertical axis) and DICE-E (horizontal axis) on a 32-node ring graph. Each node uses a 512-sample subset of CIFAR-10 or CIFAR-100. Models are trained for 5 epochs with a batch size of 16 and a learning rate of 0.1.

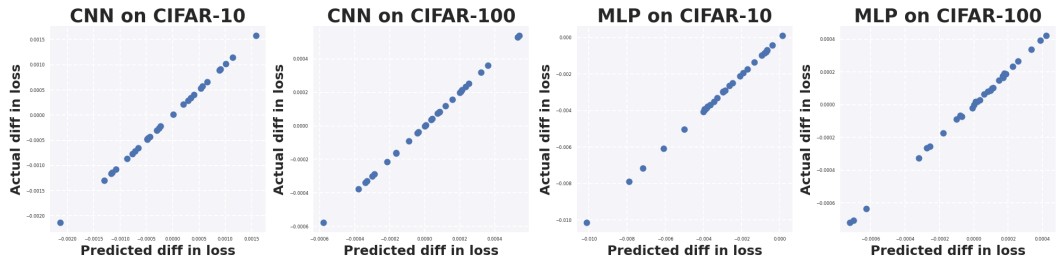

Figure D.4: Alignment between one-hop DICE-GT (vertical axis) and DICE-E (horizontal axis) on a 32-node ring graph. Each node uses a 512-sample subset of CIFAR-10 or CIFAR-100. Models are trained for 5 epochs with a batch size of 64 and a learning rate of 0.1.

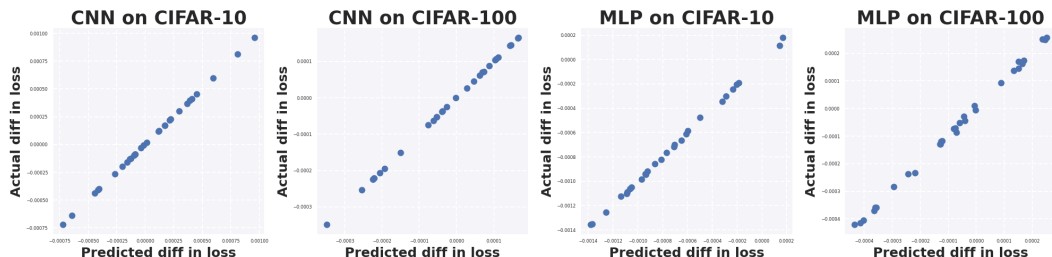

Figure D.5: Alignment between one-hop DICE-GT (vertical axis) and DICE-E (horizontal axis) on a 32-node ring graph. Each node uses a 512-sample subset of CIFAR-10 or CIFAR-100. Models are trained for 5 epochs with a batch size of 128 and a learning rate of 0.1.

### D.2.2 SENSITIVITY ANALYSIS ON LEARNING RATE AND THE NUMBER OF NODES

We also condcut experiments to evaluate the robustness of DICE-E under varying learning rates and the number of nodes.

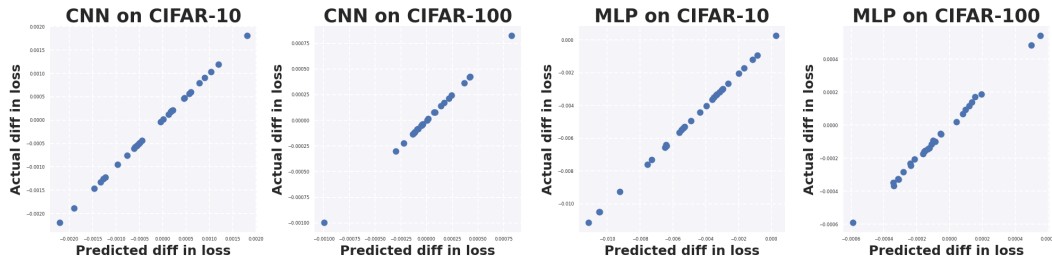

Figure D.6: Alignment between one-hop DICE-GT (vertical axis) and DICE-E (horizontal axis) on a 16-node ring graph. Each node uses a 512-sample subset of CIFAR-10 or CIFAR-100. Models are trained for 5 epochs with a batch size of 64 and a learning rate of 0.1.

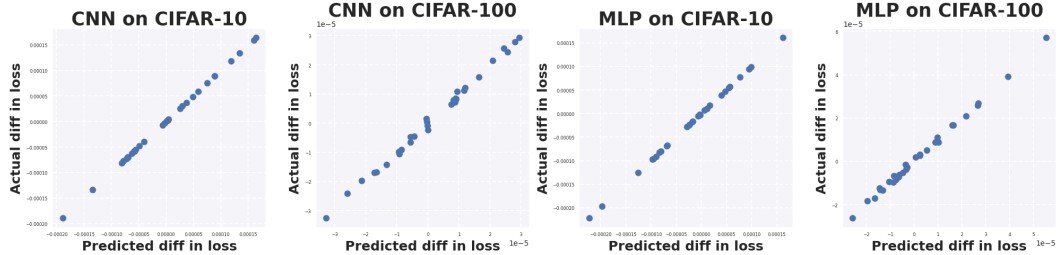

Figure D.7: Alignment between one-hop DICE-GT (vertical axis) and DICE-E (horizontal axis) on a 16-node ring graph. Each node uses a 512-sample subset of CIFAR-10 or CIFAR-100. Models are trained for 5 epochs with a batch size of 64 and a learning rate of 0.01.

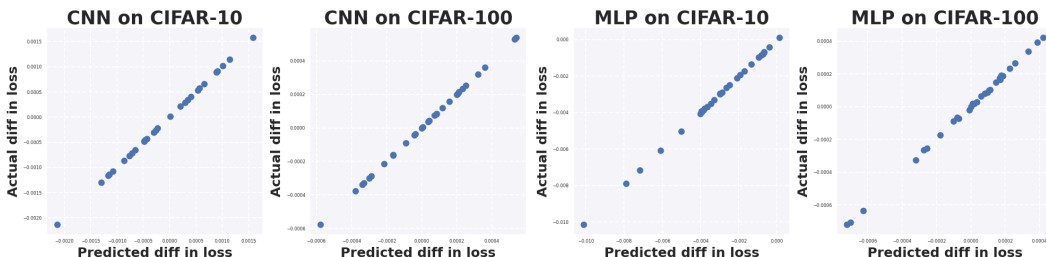

Figure D.8: Alignment between one-hop DICE-GT (vertical axis) and DICE-E (horizontal axis) on a 32-node ring graph. Each node uses a 512-sample subset of CIFAR-10 or CIFAR-100. Models are trained for 5 epochs with a batch size of 64 and a learning rate of 0.1.

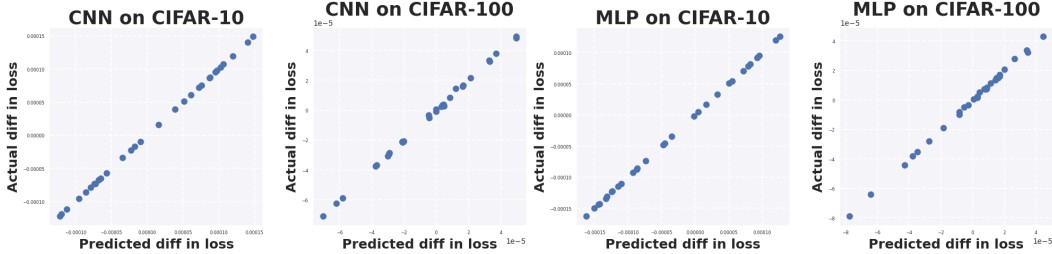

Figure D.9: Alignment between one-hop DICE-GT (vertical axis) and DICE-E (horizontal axis) on a 32-node ring graph. Each node uses a 512-sample subset of CIFAR-10 or CIFAR-100. Models are trained for 5 epochs with a batch size of 64 and a learning rate of 0.01.

### D.2.3 SENSITIVITY ANALYSIS ON TRAINING EPOCHS

We conduct experiments to evaluate the robustness of DICE-E under varying training epochs.

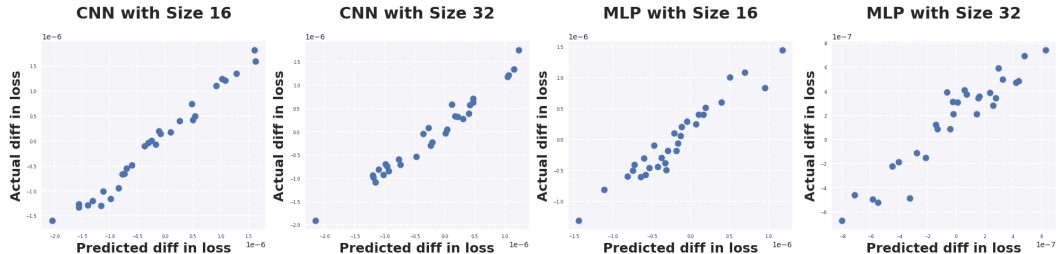

Figure D.10: Alignment between one-hop DICE-GT (vertical axis) and DICE-E (horizontal axis) on 16 and 32-node exponential graphs. Each node uses a 8192-sample subset of Tiny ImageNet. Models are trained for 10 epochs with a batch size of 128 and a learning rate of 0.1.

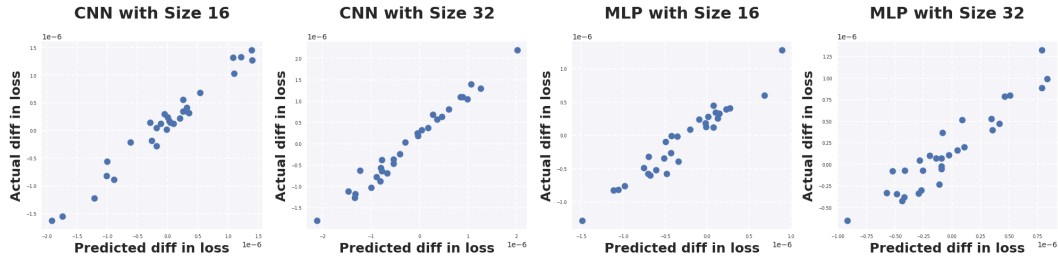

Figure D.11: Alignment between one-hop DICE-GT (vertical axis) and DICE-E (horizontal axis) on 16 and 32-node exponential graphs. Each node uses a 8192-sample subset of Tiny ImageNet. Models are trained for 20 epochs with a batch size of 128 and a learning rate of 0.1.

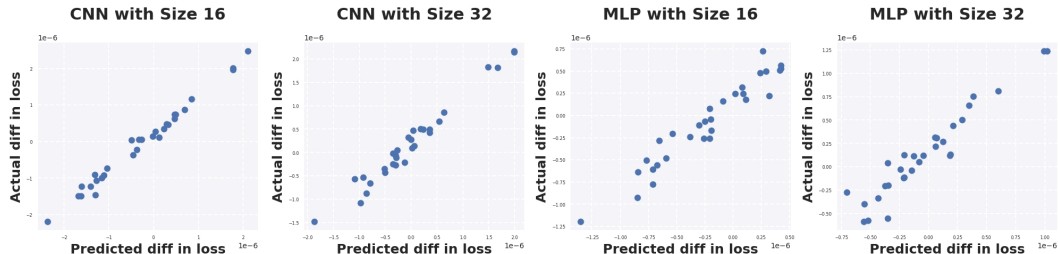

Figure D.12: Alignment between one-hop DICE-GT (vertical axis) and DICE-E (horizontal axis) on a 16 and 32-node exponential graph. Each node uses a 8192-sample subset of Tiny ImageNet. Models are trained for 30 epochs with a batch size of 128 and a learning rate of 0.1.

## D.3  ANOMALY DETECTION

We can also use the proximal influence metric to effectively detect anomalies. Specifically, anomalies are identified by observing significantly higher or lower proximal influence values compared to normal data instances. In our setup, anomalies are generated through random label flipping or by adding random Gaussian noise to features. The following Figures illustrates that the most anomalies (in red) is detectable with proximal influence values.

### D.3.1  RANDOM LABEL FLIPPING

We can conclude from these experiments that anomalies introduced through random label flipping are readily detectable by analyzing their proximal influence.

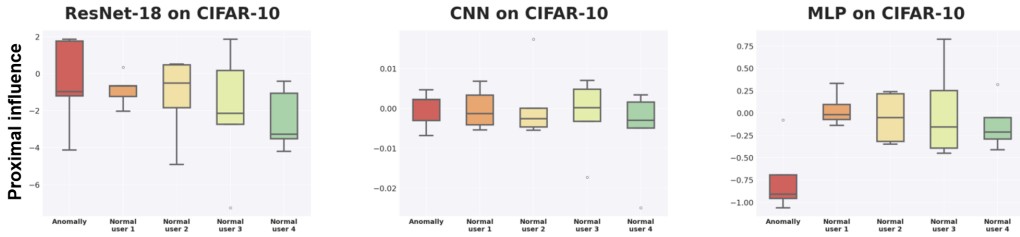

Figure D.13: Anomaly detection on exponential graph with 32 nodes. Each node uses a 512-sample subset of CIFAR-10. Models are trained for 5 epochs with a batch size of 16 and a learning rate of 0.1. In a 32-node exponential graph, each participant connects with 5 neighbors, where the neighbor in red is set as an anomaly by random label flipping, while the other four are normal participants.

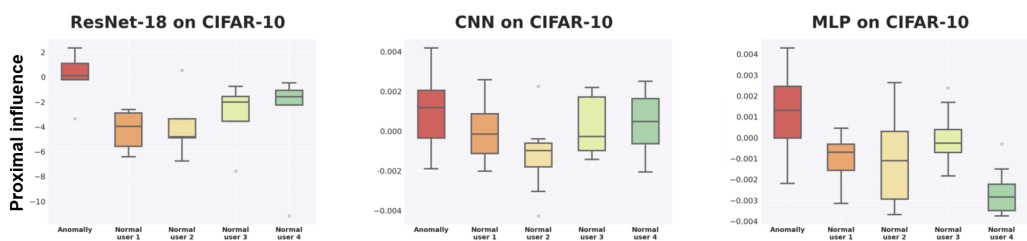

Figure D.14: Anomaly detection on exponential graph with 32 nodes. Each node uses a 512-sample subset of CIFAR-10. Models are trained for 5 epochs with a batch size of 64 and a learning rate of 0.1. In a 32-node exponential graph, each participant connects with 5 neighbors, where the neighbor in red is set as an anomaly by random label flipping, while the other four are normal participants.

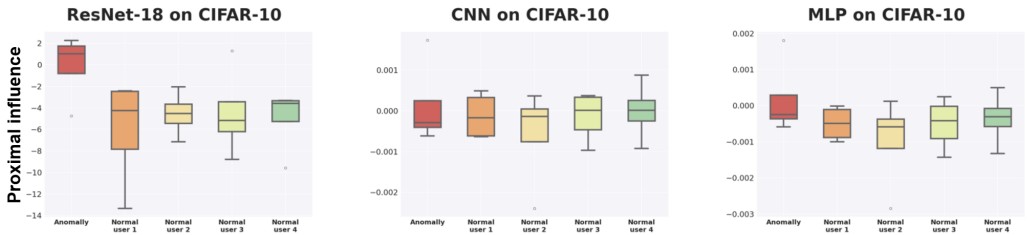

Figure D.15: Anomaly detection on exponential graph with 32 nodes. Each node uses a 512-sample subset of CIFAR-10. Models are trained for 5 epochs with a batch size of 128 and a learning rate of 0.1. In a 32-node exponential graph, each participant connects with 5 neighbors, where the neighbor in red is set as an anomaly by random label flipping, while the other four are normal participants.

### D.3.2 FEATURE PERTURBATIONS

We can conclude from Figure D.19, Figure D.20 and Figure D.21 that most anomalies introduced through adding zero-mean Gaussian noise with high variance are readily detectable by analyzing their proximal influence, which significantly deviates from that of normal data participants.

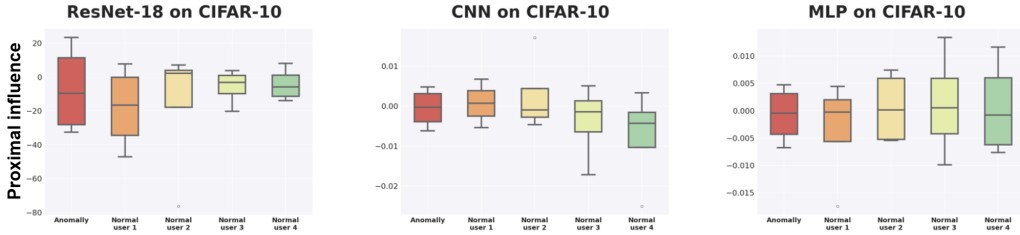

Figure D.16: Anomaly detection on exponential graph with 32 nodes. Each node uses a 512-sample subset of CIFAR-10. Models are trained for 5 epochs with a batch size of 16 and a learning rate of 0.01. In a 32-node exponential graph, each participant connects with 5 neighbors, where the neighbor in red is set as an anomaly by random label flipping, while the other four are normal participants.

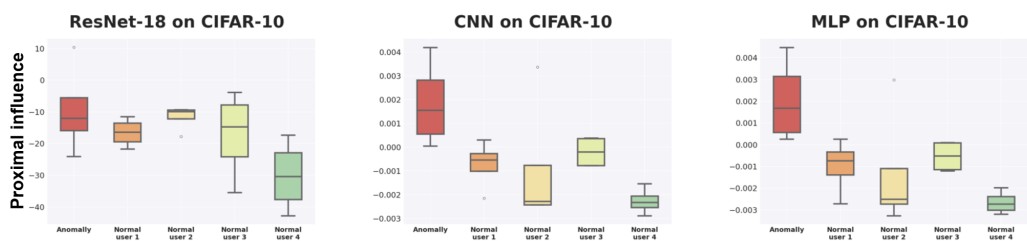

Figure D.17: Anomaly detection on exponential graph with 32 nodes. Each node uses a 512-sample subset of CIFAR-10. Models are trained for 5 epochs with a batch size of 64 and a learning rate of 0.01. In a 32-node exponential graph, each participant connects with 5 neighbors, where the neighbor in red is set as an anomaly by random label flipping, while the other four are normal participants.

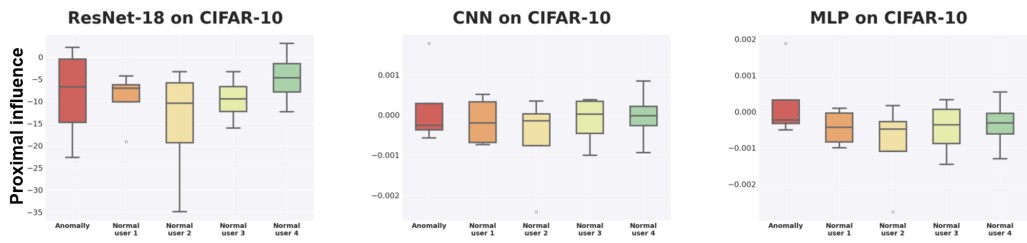

Figure D.18: Anomaly detection on exponential graph with 32 nodes. Each node uses a 512-sample subset of CIFAR-10. Models are trained for 5 epochs with a batch size of 128 and a learning rate of 0.01. In a 32-node exponential graph, each participant connects with 5 neighbors, where the neighbor in red is set as an anomaly by random label flipping, while the other four are normal participants.

## D.4 INFLUENCE CASCADE

### D.4.1 ONE-HOP INFLUENCE CASCADE

The topological dependency of DICE-E in our theory reveals the "power asymmetries" (Blau, 1964; Magee & Galinsky, 2008) in decentralized learning. To support the theoretical finding, we examine the one-hop DICE-E values of the same batch on participants with vastly different topological importance. Figure 1 illustrates the one-hop DICE-E influence scores of an identical data batch across participants during decentralized training of a ResNet-18 model on the CIFAR-10 dataset. Node sizes represent the one-hop DICE-E influence scores, quantifying how a single batch impacts other participants in the network. The dominant nodes (e.g., those with larger outgoing communication weights in $W$) exhibit significantly higher influence, as shown in Figure 1 and further detailed

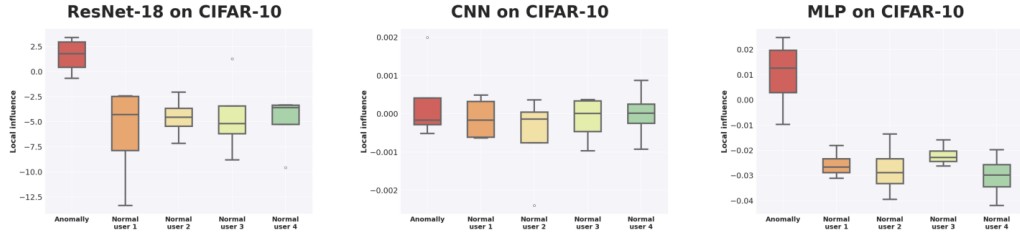

Figure D.19: Anomaly detection on exponential graph with 32 nodes. Each node uses a 512-sample subset of CIFAR-10. Models are trained for 5 epochs with a batch size of 128 and a learning rate of 0.1. In a 32-node exponential graph, each participant connects with 5 neighbors, where the neighbor in red is set as an anomaly by adding zero-mean Gaussian noise with variance equals 100 on each feature, while the other four are normal participants.

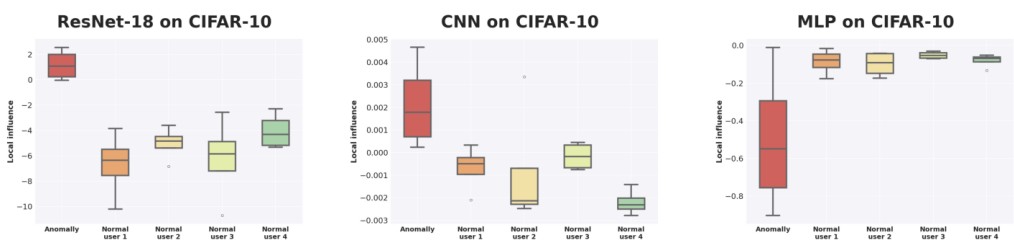

Figure D.20: Anomaly detection on exponential graph with 32 nodes. Each node uses a 512-sample subset of CIFAR-10. Models are trained for 5 epochs with a batch size of 64 and a learning rate of 0.01. In a 32-node exponential graph, each participant connects with 5 neighbors, where the neighbor in red is set as an anomaly by adding zero-mean Gaussian noise with variance equals 100 on each feature, while the other four are normal participants.

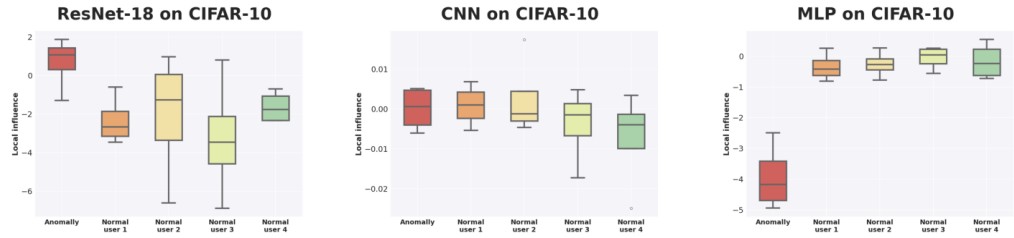

Figure D.21: Anomaly detection on exponential graph with 32 nodes. Each node uses a 512-sample subset of CIFAR-10. Models are trained for 5 epochs with a batch size of 16 and a learning rate of 0.1. In a 32-node exponential graph, each participant connects with 5 neighbors, where the neighbor in red is set as an anomaly by adding zero-mean Gaussian noise with variance equals 100 on each feature, while the other four are normal participants.

in Figure D.23 and Figure D.24. These visualizations underscore the critical role of topological properties in shaping data influence in decentralized learning, demonstrating how the structure of the communication matrix $W$ determines the asymmetries in influence.

To better observe and showcase the "influence cascade" phenomenon, we design a communication matrix with one "dominant" participant (node 0), two "subdominant" participants (nodes 7 and 10), and several other common participants. Figure D.22 (Left) visualizes the communication topology, where node sizes indicate out-degree, reflecting their influence, and edge thickness represents the strength of communication links. Node 0 stands out as the dominant participant with the largest size, while nodes 7 and 10 serve as subdominant intermediaries. Figure D.22 (Right) complements this by

showing the adjacency matrix $W$ as a heatmap, where the color intensity highlights the magnitude of connection strengths, with the dominant participant exhibiting strong outgoing links across the network. Together, these visualizations highlight the hierarchical structure and asymmetries in the communication matrix, crucial for understanding topological influences in decentralized learning.

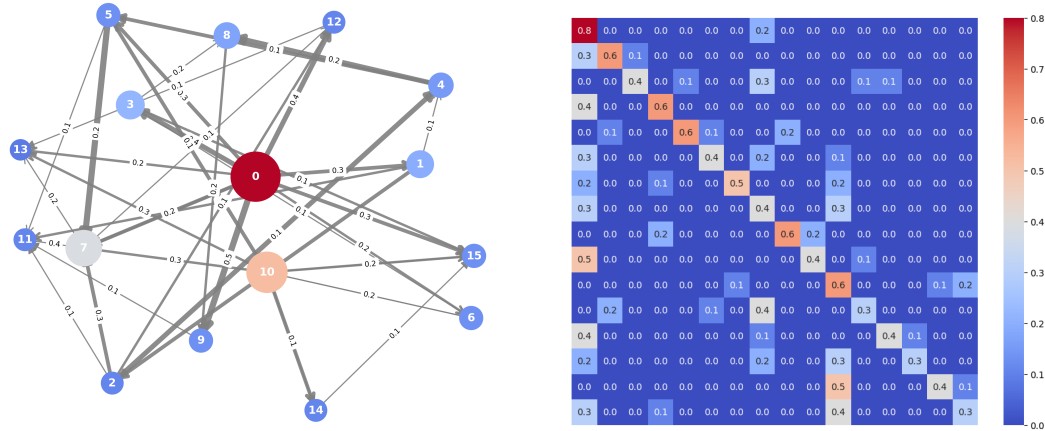

Figure D.22: **Left**: Visualization of the communication topology used in Section 5, where each node represents a participant, and edges indicate communication links. Node sizes are proportional to their out-degree (sum of outgoing edge weights), reflecting their communication influence within the community. Edge thickness corresponds to the strength of connection (i.e., weight), with directional arrows capturing the flow of information between participants. Self-loops are omitted for simplicity. **Right**: Heatmap representation of the weighted adjacency matrix $W$ used in Section 5, where each entry $W_{k,j}$ quantifies the communication strength from participant $j$ to $k$. The color intensity represents the magnitude of the weights.

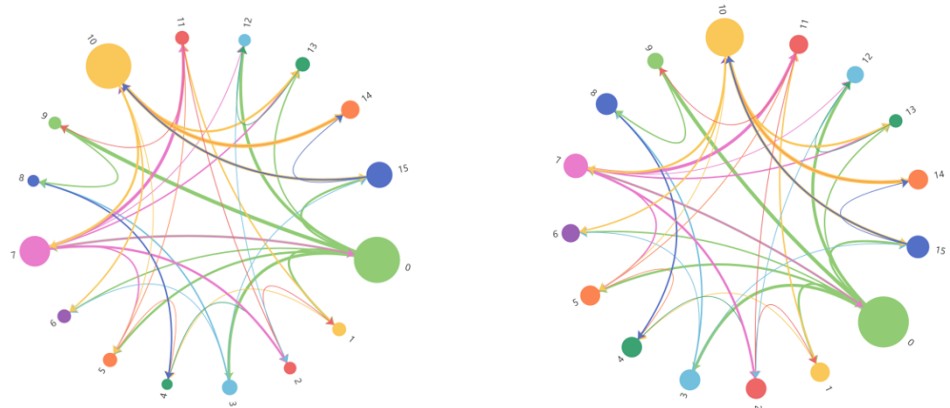

Figure D.23: Visualization of one-hop influence cascade during decentralized training with MLP on MNIST (left) and CIFAR-10 (right) under a designed communication matrix (see Figure D.22). The thickness of edges represents the strength of communication links (i.e., weights in $W$), while node sizes correspond to the relative one-hop DICE-E influence scores (see Proposition 1) computed for the same data batch across different participants. The numerical labels on the nodes indicate the corresponding participants, aligning with the participant indices in Figure D.22.

### D.4.2 MULTI-HOP INFLUENCE CASCADE

To better illustrate the communication structure underlying the *influence cascade phenomenon in multi-hop* decentralized learning (see Figure 1), following the setup in Appendix D.4.1, with the only

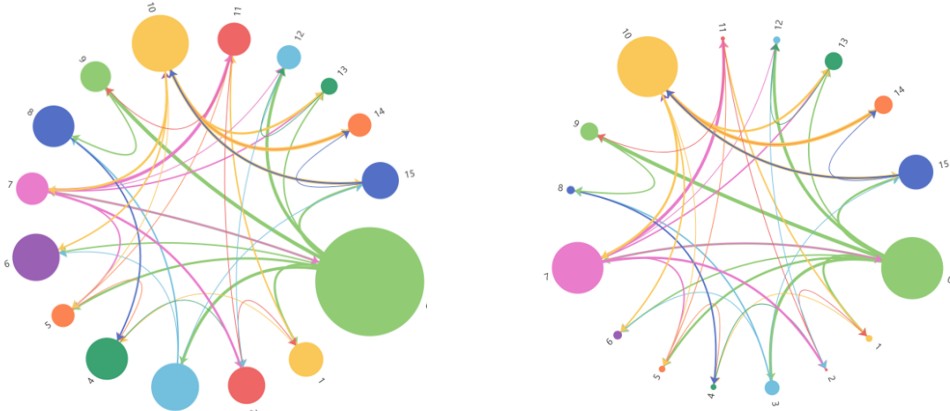

Figure D.24: Visualization of one-hop influence cascade during decentralized trainingg with ResNet-18 on CIFAR-10 (left) and CIFAR-100 (right) under a designed communication matrix (see Figure D.22). The thickness of edges represents the strength of communication links (i.e., weights in $W$), while node sizes correspond to the relative one-hop DICE-E influence scores (see Proposition 1) computed for the same data batch across different participants. The numerical labels on the nodes indicate the corresponding participants, aligning with the participant indices in Figure D.22.

modification being the use of a different mixing matrix. This modification is specifically designed to refine the visualization for better geographic representation, making the spatial relationships of decentralized participants more apparent. The heatmap in Figure D.25 visualizes the corresponding mixing matrix (i.e., weighted adjacency matrix) $W$ for the 16-node topology, where each entry $W_{j,k}$ represents the communication strength from participant $j$ to $k$. The color intensity encodes the magnitude of these weights, with warmer colors indicating stronger connections.

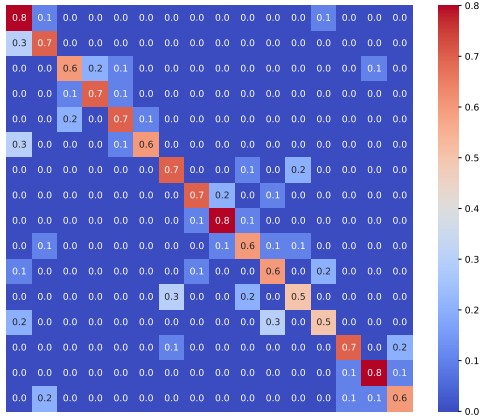

Figure D.25: Heatmap representation of the weighted adjacency matrix $W$ used in Figure 1, where each entry $W_{k,j}$ quantifies the communication strength from participant $j$ to $k$. The color intensity represents the magnitude of the weights.

