# OpenReview forum: "DICE: Data Influence Cascade in Decentralized Learning"
_ICLR.cc/2025/Conference — ICLR 2025 Poster_

### Official Review · Reviewer_4yzV · 2024-10-27

**Soundness:** 3
**Presentation:** 3
**Contribution:** 3
**Rating:** 6
**Confidence:** 3

**Summary:**

The paper proposes DICE as a framework for measuring data influence cascades in decentralized environments. The framework explains how data influence propagates through the communication network, emphasizing the interaction between the original data and the network structure in shaping data influence within decentralized learning. The experimental results show that the first-order approximation of the “gold standard” for evaluating data influence in decentralized environment can approaching the truth, and this framework can used for detecting mislabeled anomalies.

**Strengths:**

1. This paper summarizes previous work on measuring data influence and highlights the gaps in applying these methods to distributed scenarios.
2. This paper proposes a sound “gold standard” and its first-order approximation to quantify individual contributions in decentralized learning.

**Weaknesses:**

1. The experiments are weak, and Section 5.3 is unfinished.
2. The notation η^t in Theorem 1 is previously appears as η_t in Algorithm 1.

**Questions:**

Please see weaknesses.

---

> ### Author Response · Authors · 2024-11-22
> **Author Response**
>
> We thank the reviewer for helpful suggestions. We have carefully revised the manuscript to include new empirical results. Hope all your concerns are addressed.
>
> **Q1**: The experiments are weak.
>
> **A1**: In rebuttal, we strengthen our empirical results as follows:
>
> **Sensitivity Analysis**.
>
> We conduct sensitivity analysis experiments to evaluate the robustness of DICE under varying hyperparameter settings, including learning rate, batch size, and training epochs. Results demonstrate that variations in these parameters (e.g., learning rates of 0.1 and 0.01; batch sizes of 16, 64, and 128 per participant; training epochs of 5,10, 20 and 30) have minimal impact on our conclusions: (1) one-hop DICE-E provides a strong approximation of DICE-GT; (2) anomalies introduced through random label flipping and feature perturbations are readily detectable by analyzing their proximal influence. These findings highlight the robustness of DICE across diverse configurations.
>
> For your convenience, we provide an anonymous link summarizing the main experimental results:
>
> **https://anonymous.4open.science/r/Anonymous-Repo-for-Rebuttal-793D/Sensitivity%20analysis/README.md.**
>
> **Practical Applications**.
>
> DICE offers broad applicability in decentralized learning scenarios. Below, we outline two key use cases.
>
> - **Efficient Collaboration between Decentralized Participants:** DICE enables adaptive collaboration in decentralized systems by estimating the contributions of neighboring participants toward reducing its own validation loss. By leveraging this estimation, DICE facilitates dynamic reweighting strategies that adaptively prioritize interactions with more influential peers. This mechanism significantly improves both convergence speed and validation accuracy, as validated by experiments on CIFAR-10 and CIFAR-100. These results demonstrate the effectiveness of DICE in heterogeneous and decentralized learning environments.
> - **Anomaly Detection:** DICE identifies malicious neighbors, referred to as anomalies, by evaluating their proximal influence, which estimates the reduction in test loss caused by a single neighbor. A high proximal influence score indicates that a neighbor increases the test loss, negatively impacting the learning process. By detecting malicious behaviors such as label-flipping attacks or feature perturbations, DICE can enhance the reliability of decentralized learning systems.
>
> For your convenience, we provide further details and results at the following anonymous link:
>
> **https://anonymous.4open.science/r/Anonymous-Repo-for-Rebuttal-793D/Practical%20applications/README.md.**
>
> **Q2**: Section 5.3 is unfinished.
>
> **A2**: Thanks and addressed. We have carefully revised this section. The updated section highlights the purpose and significance of analyzing influence cascades, supported by detailed visual and experimental evidence.
>
> Specifically, we emphasize:
>
> - **Purpose**: Influence cascades validate our theoretical insights by illustrating how data influence propagates through network topology and reveal "power asymmetries" in decentralized learning.
> - **Findings**: As shown in Figure 1 and Appendix D.4, dominant nodes (nodes with higher outgoing communication weights in $W$) exert significantly larger influence, validating the topological dependency derived in our theory.
>
> **Q3**: The notation η^t in Theorem 1 is previously appears as η_t in Algorithm 1.
>
> **A3**: Thanks and addressed. We have carefully revised our manuscript.
>
> We have uploaded a revised version of our paper, which can be accessed via the official link provided below:
>
> **https://openreview.net/pdf?id=2TIYkqieKw**

---

> > ### Comment · Reviewer_4yzV · 2024-11-26
> > **Update score**
> >
> > I have updated the score.

---

> ### Author Response · Authors · 2024-11-26
> **Appreciating your feedback and ready to address further concerns**
>
> Thank you for updating the score! We truly appreciate your time and effort. We noticed that the updated score remains marginally below the acceptance threshold, and we would be more than happy to address any further questions or concerns you might have.

---

> > ### Comment · Reviewer_4yzV · 2024-11-26
> > **Done**
> >
> > I have updated the score.

---

> > > ### Author Response · Authors · 2024-11-26
> > > **Thank you!**
> > >
> > > Thank you so much for your kind support!

---

### Official Review · Reviewer_f3R7 · 2024-11-04

**Soundness:** 3
**Presentation:** 3
**Contribution:** 3
**Rating:** 6
**Confidence:** 4

**Summary:**

The paper proposes a method for quantifying the impact of data points in decentralized machine learning settings. The influence is measured not only at immediate neighbors but the entire network. This method can be useful for machine unlearning  or to develop new incentive mechanisms.

**Strengths:**

-  The paper is well-organized, with clear definitions, figures, and explanations that make the methods and results easy to follow.
- The paper provides a solid theoretical framework, supported by rigorous proofs and analyses.

**Weaknesses:**

- Need for more details about the practical use of this technique: While the authors use LLMs as one of the examples in the introduction, it might not be the best example to use in this case. It hard to see how this research addresses a practical problem or application that has real-world significance, or how this framework would be relevant for practitioners.
- Link with other papers that use gradient to cluster clients should be added, particularly interesting and relevant in the collaborator choice part.
- Experiments seem non-exhaustive and many details are missing to replicate the experiments. For instance, no indication on what the anomaly is vs normal client. This is particularly important when using gradients. I expect that the framework would perform differently if the anomaly is label flipping vs if it was noisy features. Additionally, evaluation of the impact of batch size would be particularly important for both scalability and compatibility among clients.

**Questions:**

1) Please motivate the approach with practical use-cases.
2) Please discuss link with clustered federated learning, in particular techniques that use gradients to cluster clients.
3) Please provide all necessary details to replicate the results.
4) Please evaluate the impact of batch size (smaller and larger values), to show the scalability of the technique and its robustness in showing the compatibility among clients.

---

> ### Author Response · Authors · 2024-11-22
> **Author Response (Part 1/4)**
>
> We thank the reviewer for the helpful comments, especially for pointing out the connections between clustered FL. We have carefully revised our manuscript in accordance with your suggestions. Hope all your concerns are cleared.
>
> **Q1**: Please motivate the approach with practical use-cases.
>
> **A1**: Thanks. DICE offers broad applicability in decentralized learning scenarios, which we summarized in Section 4.3 and Appendix B.1. For full details and results, please kindly consult the following anonymous link:
>
> **https://anonymous.4open.science/r/Anonymous-Repo-for-Rebuttal-793D/Practical%20applications/README.md**
>
> **Practical Use Case 1: Efficient Collaboration via Contribution-Based Reweighting**. It remains an open problem to set up the communication topology in decentralized learning. This challenge is largely attributed (1) local data remains private, while only parameter communication is allowed – limited insights of neighbor information are accessible, (2) the absence of a central authority to manage everything, etc. Based on DICE, we may design an adaptive topology reweighting method, as an efficient mechanism for participants adjust their collaboration strategy based on their neighbors’ contributions. to estimate their neighbors' contributions using proximal influence. Establishing an optimal communication topology in decentralized learning remains a significant open challenge. This challenge arises primarily due to:
> - The privacy-preserving nature of decentralized learning, where only model parameters are shared while local data remains private, limiting the information participants have about their neighbors.
> - The absence of a central authority, which prevents global coordination and decision-making.
>
> DICE facilitates efficient collaboration by providing a framework for participants to estimate their neighbors' contributions toward reducing their own validation loss. By leveraging this estimation, participants can adaptively reweight their gossip weights to prioritize communication with neighbors who can positively impact their learning process. DICE supports the formation of adaptive communication topologies without requiring global coordination, addressing key challenges in decentralized learning.
>
> Each participant $k$ can reduce test loss on their local dataset by minimizing the sum of proximal influences from its neighbors:
>
> $$
> I^{k,j}(z_j^t, z_k') = -\sum_{j \in N_{\text{in}}^{(1)}(k)}\eta^t  W_{k,j}^t q_k \nabla L(\theta_j^t; z_j^t)^\top \nabla L(\theta_k^{t+1}; z_k').
> $$
>
> Specifically, participant $k$ can reweight $W_{k,j}^t$ to align better with the gradient term $q_k \nabla L(\theta_j^t; z_j^t)^\top \nabla L(\theta_k^{t+1}; z_k')$, thereby reducing $I^{k,j}(z_j^t, z_k')$. A reweighting strategy can be implemented as follows:
>
> $$
> W_{k,j}^t = \frac{\nabla L(\theta_j^t; z_j^t)^\top \nabla L(\theta_k^{t+1}; z_k')}{\sum_{l \in N_{\text{in}}^{(1)}(k)} \nabla L(\theta_l^t; z_l^t)^\top \nabla L(\theta_k^{t+1}; z_k')},
> $$
>
> which ensures row-stochasticity ($\sum_{j \in N_{\text{in}}^{(1)}(k)} W_{k,j}^t = 1$).
>
> We conducted experiments to validate this algorithm, utilizing Decentralized SGD to train ResNet-18 on CIFAR-10 and CIFAR-100. The evaluation compares DICE-reweighted topologies against pre-defined topologies, such as ring and exponential configurations. To emphasize the importance of effective collaboration strategies in heterogeneous environments, we simulated data heterogeneity by partitioning the datasets using Dirichlet sampling with $\alpha = 0.3$. Each participant’s performance was measured on its local validation set, and the average validation accuracy across all participants was used as the comparison metric. The experiments were conducted with 16 participants, each utilizing a local batch size of 128 and a learning rate of 0.1. The results are summarized in the following Table and Figures.
>
> | Topology        | Merging Strategy     | CIFAR-10     | CIFAR-100     |
> | --------------- | -------------------- | ------------ | ------------- |
> | Exponential     | Fixed                | 83.83        | 53.01         |
> |                 | DICE-reweighted      | **85.53**    | **55.25**     |
> | Ring            | Fixed                | 86.92        | 56.32         |
> |                 | DICE-reweighted      | **87.21**    | **61.26**     |
>
> The experimental results demonstrate that the DICE-reweighted adaptive gossip strategy significantly outperforms the ring and exponential topologies in terms of stability, convergence speed, and validation accuracy - it has more stable training and higher validation accuracy on CIFAR-10, while exhibiting faster convergence and improved validation accuracy on CIFAR-100.

---

> ### Author Response · Authors · 2024-11-22
> **Author Response (Part 2/4)**
>
> **Practical Use Case 2: Detection of Anomalies**.
>
> DICE identifies malicious neighbors, referred to as anomalies, by evaluating their proximal influence, which estimates the reduction in test loss caused by a single neighbor. A high proximal influence score indicates that a neighbor increases the test loss, negatively impacting the learning process. By detecting malicious behaviors such as label-flipping attacks or feature perturbations, DICE can enhance the reliability of decentralized learning systems. The experimental results shows that label flipping  and feature perturbed anomalies (in red) are detectable with proximal influence values across various backbones and datasets. This application plays a critical role in addressing challenges like detecting free-riders and malicious behaviors in decentralized networks without central authorities.
>
> These practical applications align with a broader vision of DICE as a foundation for incentivized decentralized learning, facilitating the development of self-regulating  data and parameter markets.
>
> **Q2**: Please discuss link with clustered federated learning, in particular techniques that use gradients to cluster clients.
>
> **A2**: Thanks for pointing this out! There are shared aspects between gradient-based Clustered Federated Learning (CFL) and the one-hop DICE approximation, with **DICE-E potentially serving as a more advanced high-order gradient similarity metric for clustering participants in decentralized federated learning**—a promising direction for future work.  We have carefully discussed and compared them.
> Clustered Federated Learning (CFL) groups clients with similar data distributions and training collaboratively but separately within each cluster [1, 2, 3, 4]. Gradient-based CFL specifically forms these clusters using client gradient similarities [3, 4].
> For instance, [3] employs a post-convergence bi-partition to clients based on the cosine similarity of their gradients after convergence; and [4] dynamically performs spectral clustering in federated learning, leveraging gradient features as the similarity metric.
>
> The gradient-based CFL has some similarity with the “one-hop” version of DICE estimator, which both use gradient similarity information. Gradient-based CFL typically employs the cosine similarity of gradients as a clustering criterion. Similarly,  one-hop DICE-E estimates influence by considering the inner product between the training gradient of an "influence sender" and the test gradient of the evaluation node.

---

> ### Author Response · Authors · 2024-11-22
> **Author Response (Part 3/4)**
>
> We also note that there are key differences between DICE and gradient-based CFL.
>
> **Motivation**.
> - Clustered Federated Learning addresses the challenge of data heterogeneity in federated learning by grouping clients with similar data distributions, without creating personalized models for each client. Gradient-based similarity is commonly used to **cluster clients**, based on the intuition that clients with similar data-generating distributions would share similar gradients.
> - In contrast, DICE is motivated by a fundamentally different challenge: **quantifying the contributions of participants** in a decentralized learning system from a data influence perspective. Specifically, DICE provides a mechanism to measure how data at one node influences learning outcomes across the network, enabling the identification of pivotal contributors or potentially malicious actors in decentralized settings.
>
> **Theoretical Formulation**. While gradient-based CFL focuses on **peer-level gradient similarity** in one-hop, i.e., computing similarity metrics  between pairs of nodes, DICE extends this concept to evaluate **multi-hop influence propagation** in the network. DICE can systematically quantify how influence from one node diffuses across multiple intermediate nodes in the graph, incorporating factors such as network topology and optimization curvature. Mathematically, DICE generalizes gradient similarity into a **non-trivial extension for decentralized networks** by introducing the notion of $r$-hop influence, which accounts for:
> - The topological structure of the communication network.
> - The curvature information (Hessian matrices) of intermediate nodes.
> - The cascading interaction of gradients over arbitrary neighbor hops.
>
> Specifically, the $r$-hop DICE-E influence $I_{DICE-E}^{(r)}\(z_j^t, z^{\prime}\)$ is given by:
> $$
> I_{DICE-E}^{(r)}\(z_j^t, z^{\prime}\) =
> -\sum_{\rho=0}^{r} \sum_{ \(k_1, \dots, k_{\rho}\) \in P_j^{\(\rho\)} }
> \eta^{t} q_{k_\rho} \prod_{s=1}^{\rho} W_{k_s, k_{s-1}}^{t+s-1}
> \nabla L\(\theta_{k_{\rho}}^{t+\rho}; z^{\prime}\)^\top
> \prod_{s=2}^{\rho}
> \(I - \eta^{t+s-1} H\(\theta_{k_s}^{t+s-1}; z_{k_s}^{t+s-1}\)\)
> \nabla L\(\theta_{j}^{t}; z_j^t\),
> $$
> where $k_0 = j$, $P_j^{(\rho)}$ denotes the set of all sequences $k_1, \dots, k_{\rho}$ such that $k_s \in N_{out}^{(1)}(k_{s-1})$ for $s = 1, \dots, \rho$, and $H(\theta_{k_s}^{t+s}; z_{k_s}^{t+s})$ is the Hessian matrix of $L$ with respect to $\theta$, evaluated at $\theta_{k_s}^{t+s}$ and data $z_{k_s}^{t+s}$. For further details, please refer to Proposition 3.
>
> This formulation highlights a key distinction: **DICE evaluates influence across multiple hops and characterizes the interplay between data, curvature, and communication topology**—factors beyond the scope of CFL frameworks. While CFL’s gradient similarity metrics, supported by strong theoretical foundations [3, 4], effectively cluster clients, they are inherently confined to the local, peer-to-peer level similarity, making it insufficient for modeling long-range or cascading influences. DICE extends these concepts by systematically quantifying "influence cascades" through decentralized networks, providing novel insights into how data, topology, and the optimization landscape interact to shape learning outcomes.
>
> We have uploaded the revised version of our paper, which now includes the discussion on multi-hop influence on **page 8**. For your convenience, we provide the official link to the updated version of the paper below:
>
> **https://openreview.net/pdf?id=2TIYkqieKw**

---

> ### Author Response · Authors · 2024-11-22
> **Author Response (Part 4/4)**
>
> **Q3**: Please provide all necessary details to replicate the results. For instance, no indication on what the anomaly is vs normal client. Please evaluate the impact of batch size (smaller and larger values), to show the scalability of the technique and its robustness in showing the compatibility among clients.
>
> **A3**: Thanks and addressed. We have carefully provided all necessary details in Section 5 and the Appendix, including learning rate, batch size, training epochs, and different types of anomalies. To secure the reproducibility, we will release the source code package.
>
> Anomalies are generated by randomly flipping labels of training data or adding Gaussian noise to data features, please kindly refer to [5]. Furthermore, we note that varying the learning rate, batch size, training epochs, or types of anomalies, our conclusions are consistently robust, which highlights the reliability of the experimental results.
>
> For further details and results with different hyperparameter setups, please kindly consult the following anonymous link:
>
> **https://anonymous.4open.science/r/Anonymous-Repo-for-Rebuttal-793D/Sensitivity%20analysis/README.md**
>
> **Reference**
>
> [1] Three Approaches for Personalization with Applications to Federated Learning, 2020.
>
> [2] An Efficient Framework for Clustered Federated Learning, NeurIPS 2020.
>
> [3] Clustered Federated Learning: Model-Agnostic Distributed Multi-Task Optimization under Privacy Constraints. IEEE Transactions on Neural Networks and Learning Systems.
>
> [4] Clustered Federated Learning via Gradient-based Partitioning, ICML  2024.
>
> [5] Anomaly Detection and Defense Techniques in Federated Learning: A Comprehensive Review. Artificial Intelligence Review.

---

> > ### Comment · Reviewer_f3R7 · 2024-11-25
> > **Concerns addressed**
> >
> > Thank you for carefully responding to my questions. I believe this makes the paper better and the experiments reproducible.

---

> > > ### Author Response · Authors · 2024-11-26
> > > **Thank you!**
> > >
> > > Thank you for your kind feedback and support! We are delighted that all your concerns have been cleared!

---

### Official Review · Reviewer_kSES · 2024-11-05

**Soundness:** 3
**Presentation:** 3
**Contribution:** 3
**Rating:** 6
**Confidence:** 2

**Summary:**

This paper proposes the DICE framework for measuring the cascading propagation of data influence
in decentralized learning networks. Decentralized learning enables large-scale model training
through distributed computation, yet the lack of effective incentive mechanisms can lead to unfair
contributions and malicious behavior among nodes. The DICE framework introduces data influence
cascades (DICE-GT and DICE-E), which respectively measure the direct and indirect influence of data
within the network, addressing the limitations of existing data influence measurement methods in
decentralized environments. Experiments validate the consistency and accuracy of DICE across
various network topologies and demonstrate its potential in practical applications like anomaly
detection and collaborator selection

**Strengths:**

1. The DICE framework is the first to systematically measure the cascading propagation of data
influence in decentralized learning environments, providing an effective method to assess data
contributions among nodes and filling a gap in data influence evaluation within decentralized
networks.
2. The experiments cover different network topologies (such as ring and exponential graphs) and
datasets (such as MNIST, CIFAR-10, and CIFAR-100), validating the applicability and consistency
of the DICE framework across various scenarios.
3. The DICE framework provides accurate contribution measurement, laying the foundation for
designing fair and effective incentive mechanisms in decentralized learning systems, with the
potential to foster equitable collaboration within decentralized networks.

**Weaknesses:**

1. Figure 1 lacks legend information, making it difficult to understand.
2. The performance differences of the DICE framework under different parameters (such as learning
rate, batch size, etc.) have not been thoroughly discussed. It is recommended to add parameter
sensitivity experiments to demonstrate the impact of different parameter selections on the
performance of the DICE framework, thereby enhancing its practicality.

**Questions:**

See weaknesses

---

> ### Author Response · Authors · 2024-11-22
> **Author Response**
>
> We thank the reviewer for the helpful suggestions and kind support. We have carefully revised our manuscript according to your suggestions. Hope all your concerns are addressed.
>
> **Q1**: Figure 1 lacks legend information.
>
> **A1**: Thanks and addressed.  Figure 1 provides visualization of influence cascades during decentralized training with ResNet-18 on CIFAR-10 under a designed communication matrix (see details in Appendix D.4). The thickness of edges represents the strength of communication links (i.e., weights of the communication matrix), while **node sizes correspond to the one-hop DICE-E influence scores** (see Proposition 1) computed for the same data batch across different participants.
>
> **Q2**: The performance differences of the DICE framework under different parameters (such as learning rate, batch size, etc.) have not been thoroughly discussed.
>
> **A2**: Thanks and addressed. Following your suggestion, we have conducted additional sensitivity analysis experiments to evaluate the robustness of DICE under varying hyperparameters, including learning rate, batch size, and training epochs. The results are summarized in the Experiments section and Appendix D. The learning rate is set as 0.1 or 0.01, batch size is set as 16, 64, or 128 per participant, training epoch is set as 5, 10, 20 and 30. Our conclusions robustly stand in all the settings: (1) One-hop DICE-E provides a strong approximation of DICE-GT; (2) anomalies introduced through random label flipping and feature perturbations are readily detectable by analyzing their proximal influence.This demonstrates the stability and robustness of the DICE approximation across different setups.
>
> For further details and results, please kindly consult the following anonymous link:
>
> **https://anonymous.4open.science/r/Anonymous-Repo-for-Rebuttal-793D/Sensitivity%20analysis/README.md**

---

> > ### Comment · Reviewer_kSES · 2024-11-25
> > **Acknowledge the rebuttal**
> >
> > Thank you for your feedback. After carefully considering your rebuttal, I believe the current score accurately reflects the work's strengths and areas for improvement. I would like to maintain the current score based on this evaluation.

---

> > > ### Author Response · Authors · 2024-11-25
> > > **Thank you!**
> > >
> > > Thank you for your kind support!

---

### Author Response · Authors · 2024-11-22
**General Response (Part 2/2)**

**Clarifications of Challenges and Contributions**
- **Major challenges**. Decentralized learning emerges as a promising paradigm to accommodate the growing demand for distributed computation, while also introduces distinct and unparallel challenges in quantifying and understanding data influence. In a centralized learning system, data influence is typically confined to a single model and can be statically analyzed after training. In decentralized learning, the influence of a single data instance propagates from directly connected neighbors to neighbors faraway. This put major challenge:  **`measuring multi-hop influence beyond one-hop influence, which is determined not only by the stem node, but also the nodes on the way.`** The challenge invalidate the existing influence measure techniques. This phenomenon is termed as the **cascading effect** in our paper.
- **Major contributions**. Our **DICE (Data Influence CascadE)** method is the first work in the literature that can address this challenge. DICE introduces the concept of ground-truth data influence for decentralized learning, seamlessly integrating direct and indirect contributions to capture influence propagation across multiple hops during training. By transforming data-level influence into model-level influence and tracing the model-level influence of multi-hop neighbors, we developed a theoretical framework to derive tractable approximations for influence cascades. **`We uncover, for the first time, that data influence in decentralized learning is shaped by a synergistic interplay of  original data, the topological importance of the data owner, and the curvature information of intermediate nodes mediating propagation.`** These theoretical insights enable a systematic and interpretable understanding of decentralized data influence, laying the groundwork for incentivized collaboration, anomaly detection, and scalable decentralized learning ecosystems.

---

### Author Response · Authors · 2024-11-22
**General Response (Part 1/2)**

Dear AC and Reviewers:

We sincerely thank the reviewers for their helpful comments. We appreciate that reviewers recognize the solidity of our theoretical framework (Reviewers kSES and f3R7), especially for proposing the  “gold standard” data influence measures in decentralized learning and its first-order approximation (Reviewer 4yzV).

We have carefully revised our manuscript accordingly. The main revisions and contributions are highlighted below:

**Highlight for Revision**
1. We conducted **additional experiments** following reviewers’ feedback:
    - Comprehensive sensitivity analysis on hyperparameters (including batch size, learning rate, training epoch, communication topology and the number of participants) to evaluate the robustness of DICE (to address Reviewers kSES and f3R7).
    - Additional experiments on Tiny ImageNet (to address Reviewer 4yzV). To secure the reproducibility, we have also provided detailed replication instructions in our revised manuscript, including the definition of anomalies methodologies(to address Reviewer f3R7).
2. We discussed **practical use cases of DICE** (to address Reviewer f3R7).
    - **Efficient collaboration between decentralized participants:** DICE enables estimation of the contributions between neighboring participants. Leveraging this, DICE make dynamic reweighting strategies possible, which adaptively prioritizes interactions with more influential peers. This mechanism can significantly improve both convergence speed and validation accuracy, as validated by experiments on CIFAR-10 and CIFAR-100. These results demonstrate the effectiveness of DICE in heterogeneous and decentralized learning environments.
    - **Anomaly Detection:** DICE strengthens the robustness of decentralized networks by identifying anomalies, such as label-flipping attacks or feature perturbations, by observing the deviations in proximal influence values. This capability is critical for detecting free riders, mitigating malicious behaviors, and maintaining system reliability, even under constraints of limited communication.
3. We added a comprehensive **discussion of related works on clustered federated learning** (to address Reviewer f3R7).
    - There are shared aspects between gradient-based Clustered Federated Learning (CFL) and the one-hop DICE approximation, with DICE-E potentially serving as a more advanced high-order gradient similarity metric for clustering participants in decentralized federated learning—a promising direction for future work.  We have carefully discussed and compared them. We also note that there are **key differences** between DICE and gradient-based CFL. While CFL is confined to  peer-level clustering, which is in **one-hop**, DICE non-trivially extend the influence to measuring the **multi-hop influence propagation** to whole decentralized networks.   Specifically, the r-hop DICE-E influence $I_{DICE-E}^{(r)}(z_j^t, z^{\prime})$ is given by:
$
I_{DICE-E}^{(r)}\(z_j^t, z^{\prime}\) =
-\sum_{\rho=0}^{r} \sum_{ \(k_1, \dots, k_{\rho}\) \in P_j^{\(\rho\)} }
\eta^{t} q_{k_\rho} \prod_{s=1}^{\rho} W_{k_s, k_{s-1}}^{t+s-1}
\nabla L\(\theta_{k_{\rho}}^{t+\rho}; z^{\prime}\)^\top
\prod_{s=2}^{\rho}
\(I - \eta^{t+s-1} H\(\theta_{k_s}^{t+s-1}; z_{k_s}^{t+s-1}\)\)
\nabla L\(\theta_{j}^{t}; z_j^t\),
$
where $k_0 = j$, $P_j^{(\rho)}$ denotes the set of all sequences $k_1, \dots, k_{\rho}$ such that $k_s \in N_{out}^{(1)}(k_{s-1})$ for $s = 1, \dots, \rho$, and $H(\theta_{k_s}^{t+s}; z_{k_s}^{t+s})$ is the Hessian matrix of $L$ with respect to $\theta$, evaluated at $\theta_{k_s}^{t+s}$ and data $z_{k_s}^{t+s}$.
For further details, please refer to Proposition 3. This formulation highlights a key distinction: **DICE evaluates influence across multiple hops and characterizes the interplay between data, curvature, and communication topology**—factors beyond the scope of CFL frameworks., which primarily aims to cluster clients based on their similarity.

For your convenience, we provide an anonymous link summarizing our additional experimental results:

**https://anonymous.4open.science/r/Anonymous-Repo-for-Rebuttal-793D/README.md**

We have uploaded a revised version of our paper, which can be accessed via the official link provided below:

**https://openreview.net/pdf?id=2TIYkqieKw**

Best regards,

The Authors

---

### Meta-Review · Area_Chair_eVLp · 2024-12-08

**Metareview:**

This paper introduces the DICE framework, a systematic approach to measure cascading data influence in decentralized learning networks, addressing a critical gap in data contribution evaluation. With rigorous theoretical foundations and diverse experimental validations, it lays the groundwork for equitable incentive mechanisms and effective collaboration in decentralized systems. There were concerns in the paper on practical use cases, experiments, and related work in the original reviews which seem to have been addressed in the rebuttal. Further, given that all the reviews are positive after rebuttal, I recommend acceptance of this work.

**Additional Comments On Reviewer Discussion:**

NA

---

### Decision · Program_Chairs · 2025-01-22

Accept (Poster)